# MIMIC: Mask-Injected Manipulation Video Generation with Interaction Control

**Tianxiao Chen**[1]    **Jintao Rong**[2]    **Huajin Chen**[1]
**Jingya Wang**[3]    **Tao Zhou**[1]    **Jiming Chen**[1]    **Qi Ye**[1,4†]
[1]Zhejiang University    [2]Zhejiang University of Technology    [3]ShanghaiTech University
[4]Zhejiang Key Laboratory of Airspace Awareness and Autonomous Unmanned Systems

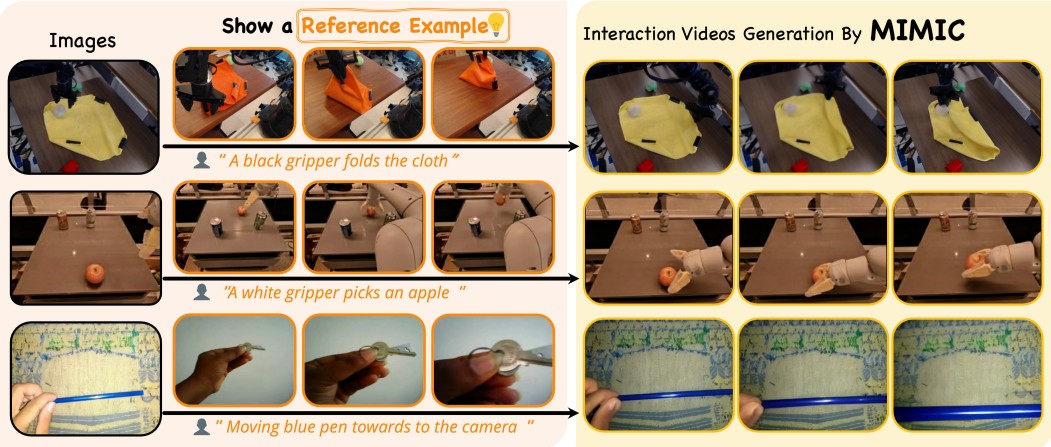

Figure 1: We propose MIMIC, a novel approach for video generation in manipulation scenarios. Given a reference video, MIMIC conditions on it to generate a new video that preserves the same operational semantic information.

## Abstract

Embodied intelligence faces a fundamental bottleneck from limited large-scale interaction data. Video generation offers a scalable alternative, but generating manipulation videos remain particularly challenging, as they require capturing subtle, contact-rich dynamics. Despite recent advances, video diffusion models still struggle to balance semantic understanding with fine-grained visual details, restricting their effectiveness in manipulation scenarios. Our key insight is that reference videos provide rich semantic and motion cues that can effectively drive manipulation video generation. Building on this, we propose MIMIC, a two-stage image-to-video diffusion framework: (1) we first introduce an Interaction-Motion-Aware (IMA) module to fuse visual features from the reference video, producing coherent semantic masks that correspond to the target image, (2) then utilize these masks as control signals to guide the video generation process. Considering the ambiguity of the motion attribution, we further introduce a Pair Prompt Control mechanism to disentangle object and camera motion by adding the reference video as an additional input. Extensive experiments demonstrate that MIMIC significantly outperforms existing methods, effectively preserving manipulation intent and motion details, even when handling diverse and deformable objects. Our findings underscore the effectiveness of reference-driven semantics for controllable and realistic manipulation video generation.

## 1 Introduction

Embodied intelligence has made notable progress Black et al.; Bjorck et al. (2025); Agarwal et al. (2025), but its development is still hindered by the scarcity of large-scale, high-quality interaction

† Corresponding author.

data . A promising alternative is to learn from manipulation videos, which naturally encode rich interaction cues and can provide valuable guidance for embodied agents Lum et al. (2025). Building on this idea, video generation He et al. (2022); Wan et al. (2025) offers a scalable solution by not only leveraging existing videos but also simulating realistic new ones, thereby augmenting training data and promoting more generalizable robot learning. Generative models in this setting can capture intrinsic video patterns and synthesize target scenes from language descriptions, *e.g.* "a person folding clothes at home". Yet, language alone is insufficient for manipulation scenarios, where subtle motions and contact dynamics must be faithfully represented.

Generating realistic manipulation videos is particularly challenging because they involve complex, contact-rich interactions between hands (or grippers) and objects. Although recent video diffusion models Xing et al. (2024); Guo et al. (2023); Yang et al. (2024) have advanced rapidly, they still struggle to balance abstract semantic understanding with fine-grained visual detail Tan et al. (2025), making it difficult to capture the nuances of manipulation behaviors. Demonstrations, in contrast, naturally convey both high-level semantics (*e.g.* folding clothes) and fine-grained interaction cues. This observation inspires our key idea: **Show a Reference Example to the Model**—that is, guiding diffusion models with a reference video alongside textual descriptions to generate manipulation sequences.

Existing general-purpose methods incorporate additional control signals, such as drag points Yin et al. (2023), object depth Xu et al. (2024b), or hand meshes Fan et al. (2025), to explicitly constrain motions. While effective for certain tasks, such strong constraints often reduce flexibility and may even produce physically implausible results. FlexiAct Zhang et al. (2025) adopts a reference video strategy similar to ours and extracts global motion representations from the reference video for general video generation. However, when applied to manipulation generation, these methods struggle to effectively handle manipulation scenarios due to complex motions between multiple objects. In addition, the generated videos demonstrate scale inaccuracies and incorrect modeling of interactions as (1) the reference and target scenes commonly exhibit large misalignments in manipulated objects, initial poses, and background contexts; and (2) the model often produces physically implausible motions, since it strictly follows the imposed control signals while neglecting the underlying causal dependencies of real-world interactions. These limitations highlight the intrinsic difficulty of manipulation generation, as success demands both structural alignment across scenes and explicit reasoning about the physical dynamics of interactions.

To address the above challenges, we propose a novel manipulation generation framework **MIMIC** by opening the black box of single-stage generation and explicitly inject the capability of manipulation-centric understanding, improving interpretability and controllability. Specifically, given a reference video, the first frame of the target scene, and a textual description, we decompose generation into two stages. The *first stage* is trained to jointly identify the object to be manipulated in the target initial frame and synthesize a temporally-coherent, physically-plausible interaction motion trajectory, which is represented as a sequence of masks. A novel **Interaction-Motion-Aware (IMA)** layer is proposed to embed interaction semantics to guide subsequent video synthesis by learning **IMA** embeddings from visual and mask embeddings from reference videos and injecting the embeddings into the generation process via **IMA** attention.

Given the generated mask sequence and the target initial image, the *second stage* renders the final realistic video. We observe that using a single mask as the control signal couples object motion with camera motion due to the lack of background information, which makes it challenging to capture interactive motions within the video. To accommodate scenarios with possible camera motion, we introduce a **Pair Prompt Control** mechanism that conditions the rendering stage on both the predicted interaction mask and the original reference video. This approach resolves the inherent ambiguity in mask-based control, allowing the model to disentangle object motion from camera motion and generate temporally coherent videos that respect the global scene dynamics.

We construct a dedicated benchmark for video generation of manipulation scenarios that includes human hands and grippers to evaluate the performance of MIMIC. Experiments demonstrate that our method effectively transfers manipulation information from the reference video to the generated video, demonstrating both **plausible motion patterns** and the **fine visual details of video**. As shown in Fig. 1, MIMIC exhibits strong capabilities across diverse manipulation scenarios and maintains high-quality generation even when handling deformable objects.

Our key contributions are summarized as follows:

- We propose **MIMIC**, a novel image-to-video diffusion framework tailored for manipulation scenarios, which leverages semantic extraction from reference videos combined with explicit interaction masks to generate physically plausible and semantically accurate manipulation videos.

- We design an **Interaction-Motion-Aware** attention mechanism that effectively embed latent manipulation semantics into the video generation process, addressing the challenges of complex motion representation without relying on predefined control signals.

- We propose a **Pair Prompt Control** mechanism that integrates reference videos with predicted interaction masks, enabling the model to effectively incorporate semantic information and reduce ambiguities inherent in mask-based control, thereby enhancing the coherence and realism of generated manipulation videos.

## 2 RELATED WORK

### 2.1 IMAGE-TO-VIDEO DIFFUSION MODELS

Leveraging the advantages of diffusion models Ho et al. (2020); Song et al. (2020), the field of image generation has witnessed remarkable advancements in content creation Rombach et al. (2022); Nichol et al. (2021); Betker et al. (2023). Following this success, video diffusion He et al. (2022) rapidly evolves by integrating high-fidelity image priors with temporal modeling to synthesize coherent short videos. Early Image-to-Video (I2V) diffusion work begins with AnimateDiff Guo et al. (2023), which introduces a lightweight Motion Adapter and MotionLoRA to animate Stable Diffusion Rombach et al. (2022), followed by SVD Blattmann et al. (2023), which augments image-conditioned diffusion with temporal convolutions and inter-frame attention. Building on these UNet-based methods, DynamiCrafter Xing et al. (2024) employs a dual-stream injection of visual detail and CLIP-aligned context to animate open-domain images into videos. More recently, DiT-based architectures Peebles & Xie (2023) such as CogVideoX Yang et al. (2024) combine a 3D VAE with an Expert Transformer to generate high-fidelity video, and Wan2.1 Wan et al. (2025) fuses a causal 3D VAE with a Diffusion Transformer and shared MLP temporal embeddings to scale generation to arbitrary lengths. We inherit the I2V diffusion backbone and its strong video priors, but apply it to manipulation by explicitly modeling hand–object interactions.

### 2.2 VIDEO MOTION CUSTOMIZATION

Motion customization seeks to generate videos that replicate specific motion patterns from reference videos while aligning with textual semantics. Early works such as Tune-A-Video Wu et al. (2023) enable one-shot video generation by fine-tuning Stable Diffusion Rombach et al. (2022). ControlVideo Zhao et al. (2025b) and Text2Video-Zero Khachatryan et al. (2023) extend Control-Net Zhang et al. (2023) with cross-frame interactions for zero-shot controllable video synthesis, while Control-A-Video Chen et al. (2023) incorporates trainable motion layers to model temporal dynamics conditioned on diverse cues. To better decouple appearance and motion, methods like VMC Jeong et al. (2024), MotionDirector Zhao et al. (2024), and MotionInversion Wang et al. (2024) train motion-specific modules that generalize across scenes and prompts. FlexiAct Zhang et al. (2025) enables one-shot complex action transfer by combining spatial adapters and a trainable frequency-aware embedding. Training-free strategies, including DMT Yatim et al. (2024) and MotionClone Ling et al. (2024), extract motion priors from latents for inference-time customization. Unlike the above approaches, we employ an in-context video generation paradigm that leverages motion information from a reference video to produce the corresponding manipulation videos.

### 2.3 INTERACTION VIDEO GENERATION

Generating interactive manipulation videos faces significant challenges, including ensuring physical plausibility and handling occlusions among multiple objects. To address these issues, some methods Xu et al. (2024b); Pang et al. (2025); Fan et al. (2025) incorporate fine-grained control signals to specify detailed manipulation processes. For instance, AnchorCrafter Xu et al. (2024b) uses hand meshes and object depth maps as inputs, while Re-Hold Fan et al. (2025) describes interactions

through hand-object bounding boxes. Other approaches aim for the model to implicitly learn physical motion patterns. CosHand Sudhakar et al. (2024) and InterDyn Akkerman et al. (2025) utilize explicit hand masks as control signals to guide the learning of object motions influenced by hand movements, and Taste-Rob Zhao et al. (2025a) similarly controls via hand keypoints. Beyond direct video generation, recent work has also explored inpainting-based editing hand–object interaction content. AffordanceDiffusion Ye et al. (2023) edits object-only images by inserting plausible hand configurations to create HOI scenes, while HOI-Swap Xue et al. (2024) and Re-Hold Fan et al. (2025) edits HOI videos by replacing the interacted object while keeping the hand motion intact. These methods focus on modifying existing content rather than generating novel interaction videos from scratch. Furthermore, recognizing the difficulty of directly generating manipulation videos with diffusion models, several methods adopt a coarse-to-fine learning strategy by first predicting a coarse control signal, then generating videos conditioned on it. For example, Img2Flow2Act Xu et al. (2024a) generates object motion trajectories via diffusion models, and FLIP Gao et al. (2024) performs uniform pixel-space sampling and predicts trajectories for each point. Existing methods rely on a single control signal to constrain the limited motions of generated videos. In contrast, our approach decouples multiple motion features from reference videos as control signals, thus avoiding complicated inputs while accurately capturing the manipulation process.

## 3 PRELIMINARY

Our method is built upon the image-to-Video model DynamiCrafter Xing et al. (2024), which mainly comprises a diffusion UNet Ronneberger et al. (2015) $\epsilon_\theta$ with spatial and temporal layers, and a variational autoencoder(VAE) Kingma & Welling (2013) composed of an encoder $\mathcal{E}(\cdot)$ and a decoder $\mathcal{D}(\cdot)$. During training, a video with $F$ frames $V \rightarrow x^{1:F} \in \mathbb{R}^{F \times 3 \times H \times W}$ is encoded into latent space as $z_0^{1:F} = \mathcal{E}(x^{1:F}) \in \mathbb{R}^{F \times d \times h \times w}$.

The forward diffusion process corrupts the video via

$$z_t^{1:F} = \sqrt{\bar{\alpha}_t}\, z_{t-1}^{1:F} + \sqrt{1 - \bar{\alpha}_t}\, \epsilon_t, \quad \epsilon_t \sim \mathcal{N}(0, \mathbf{I}), \tag{1}$$

where $t \in \{1, \dots, T\}$ indexes time steps and $\bar{\alpha}_t$ controls the noise scale. The reverse process is modeled by the diffusion UNet $\epsilon_\theta$, which estimates the noise given a noisy latent, conditioning on the initial image of the target scene $I_{tar}$ and the textual description $c$. Diffusion model is trained using the following loss:

$$\mathcal{L}_{diff} = \mathbb{E}_{z_0^{1:F}, c, \epsilon_t, t} \big\| \epsilon_t - \epsilon_\theta(z_t^{1:F}, c, I_{tar}, t) \big\|_2^2. \tag{2}$$

During inference, starting from a Gaussian noise $\epsilon$, the model iteratively denoises with $T$ timesteps to obtain an estimated clean latent $\hat{z}_0^{1:F}$, which is then decoded to the realistic video by $\mathcal{D}(\cdot)$.

## 4 METHODOLOGY

### 4.1 OVERVIEW OF MIMIC

As shown in Fig. 2, we suggest an in-context video-generation paradigm: conditioned on a reference manipulation video $V_{ref}$, an initial image $I_{tar}$ of the target scene, and a textual description $c$, the model is required to produce a corresponding manipulation video $V_{tar}$ in the target environment. The generation process is explicitly divided into two stages to promote the understanding of interactive dynamics and enhance the controllability of the manipulated object state. The first stage aims at jointly identifying the manipulated object in the target initial frame $I_{tar}$ and synthesizing a temporally-coherent, physically-plausible motion trajectory. The trajectory is represented by a sequence of masks $\mathcal{M}_{tar}$ to achieve pixel-level control of the state of the object while accommodating possible non-rigid deformations. Given the predicted mask sequence $\mathcal{M}_{tar}$ and the target initial image $I_{tar}$, the second stage renders the final realistic video, where we propose a Pair Prompt Control mechanism to accommodate scenarios with potential camera motion.

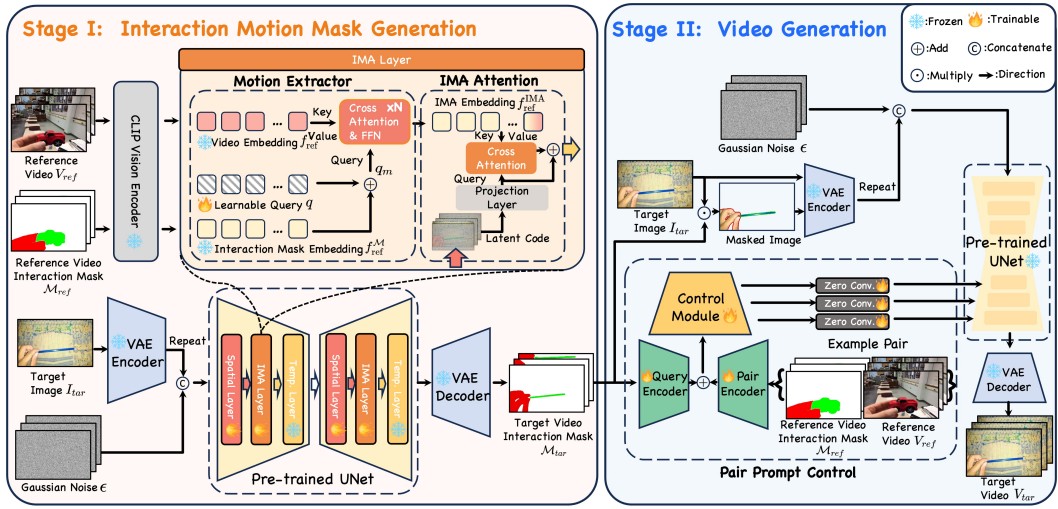

Figure 2: **Pipeline of MIMIC.** (1) **Stage I** illustrates interaction motion mask generation. We construct a Motion Extractor to capture the interaction motion information from the reference video. And then we utilize a transformer layer after the spatial layer to inject this motion information into the UNet by **Interaction-Motion-Aware (IMA) attention**. (2) **Stage II** illustrates video generation with interaction masks. We integrate a set of example pairs, each consisting of a reference video and its corresponding interaction mask, with the predicted interaction mask of the target video through the **Pair Prompt Control** module. This integration is injected into the UNet to facilitate video generation.

## 4.2 STAGE I: JOINT PERCEPTION AND INTERACTION MOTION GENERATION

Given a reference manipulation video $V_{ref}$, reference manipulation masks $\mathcal{M}_{ref}$ [1], and a textual description $c$, Stage I is explicitly trained to jointly recognize the manipulated object in the target initial frame $I_{tar}$ and synthesize a temporally-coherent, physically-plausible motion trajectory represented by masks $\mathcal{M}_{tar}$.

To fully leverage the interactive semantic information and motion information contained in the reference video $V_{ref}$, we introduce reference video embeddings $f_{ref}^V$ and interaction mask embeddings $f_{ref}^{\mathcal{M}}$, which are respectively extracted from the reference video $V_{ref}$ and the corresponding interaction mask $\mathcal{M}_{ref}$ by a frozen CLIP Radford et al. (2021) visual encoder $\Phi$. Complementary to these semantic embeddings, a lightweight **Motion Extractor** is further employed to inject reference motion cues into the denoising U-Net. To further enhance the alignment of interactive semantics and motion information, we fuse a learnable query embedding $q \in \mathbb{R}^{F \times n \times d}$ with interaction mask embedding $f_{ref}^{\mathcal{M}}$ via element-wise addition, yielding an accumulated query $q_m = q + f_{ref}^{\mathcal{M}}$ as the input to the extractor. The accumulated query interacts with the frozen video embeddings through a cross-attention layer (CA), followed by a feedforward network (FFN), obtaining the **Interaction-Motion-Aware(IMA)** embedding $f_{ref}^{IMA}$.

$$f_{ref}^{IMA} = \text{FFN}(\text{CA}(q_m, f_{ref}^V, f_{ref}^V)). \tag{3}$$

This IMA embedding $f_{ref}^{IMA}$ then interacts with the diffusion model via another cross-attention layer, so that the diffusion process is guided by the understanding of manipulation semantics. To maintain the stability of training, the output projection layer inside this attention layer is zero-initialized and equipped with a residual connection.

For the training strategy, we train the **Stage I** model in a two-step manner. Firstly, we construct static videos with repeated first frames, allowing the model to focus exclusively on learning to recognize the hand-object interactions occurring in the first frame of the target scene. Subsequently, we restore temporal dynamics and train the model to generate manipulation motions. Both phases optimize the objective in Eq. 2.

---

[1]For data without annotated masks, we utilize Grounding-SAM2 Ren et al. with language inputs to generate corresponding masks.

## 4.3 STAGE II: VIDEO GENERATION WITH INTERACTION MASKS

The objective of **Stage II** is to generate a temporally coherent and detail-rich video from the interaction masks $\mathcal{M}_{tar}$ predicted in **Stage I**. Mask-only conditioning is inherently ambiguous: it specifies where an interaction occurs but cannot disambiguate object versus camera motion or capture how the manipulation unfolds. In manipulation-centric scenarios, this limitation often results in weak consistency along the interaction trajectory and unrealistic rendering of hands or grippers.

To address these issues, we propose **Pair Prompt Control**, which conditions generation on both the target mask sequence $\mathcal{M}_{tar}$ and a reference pair $\mathcal{M}_{ref}, V_{ref}$. While the target mask provides spatial alignment, the reference pair contributes semantic and motion priors, thereby reducing mask ambiguity and enabling manipulation-aware synthesis. Architecturally, we adopt a ControlNet-style Zhang et al. (2023) control branch, where lightweight convolutional *Query Encoder* and *Pair Encoder* modules are used to process the target mask sequence $\mathcal{M}_{tar}$ and the example pair $\mathcal{M}_{ref}, V_{ref}$, respectively. The encoded features are fused within a control module, which then injects multi-scale guidance into the UNet backbone, ensuring reference-driven conditioning throughout the generation process.

To enhance fidelity and consistency within interaction regions, we use the **Stage I** predicted mask sequence $\mathcal{M}_{tar}$ together with the target image $I_{tar}$ to form a masked image $I_{\text{masked}} = I_{tar}^1 \odot m_{tar}^1$ that preserves only the interaction areas. This masked image is concatenated with the original target image as input to the diffusion model, providing explicit appearance guidance. Additionally, we reweight the diffusion loss $\mathcal{L}_{diff}$ with an adaptive region loss that emphasizes mask-aligned areas across time by combining the current interaction mask $m_{tar}^f, f = 1, ..., F$ and the first-frame mask $m_{tar}^1$:

$$\mathcal{L}_{region} = \Big( \frac{S}{S_{\mathcal{M}_{tar}}} \mathcal{M}_{tar} + \frac{S}{S_{\mathcal{M}_{tar}^1}} \mathcal{M}_{tar}^1 \Big) \odot \mathcal{L}_{diff},$$
$$\mathcal{M}_{tar}^1 = \text{Repeat}(m_{tar}^1, F),$$

(4)

where $\text{Repeat}(x, n)$ repeats $x$ along the temporal dimension, and $S_{\mathcal{M}_{tar}}, S_{\mathcal{M}_{tar}^1}$ denote the corresponding mask areas. The final training objective is defined as:

$$\mathcal{L}_{final} = (1 - \mathcal{M}_{tar} - \mathcal{M}_{tar}^1) \odot \mathcal{L}_{diff} + \lambda \mathcal{L}_{region}.$$

(5)

By combining masked-image conditioning with adaptive region loss, the model focuses learning on the relevant regions, reducing ghosting artifacts and improving both visual fidelity and temporal consistency of generated videos.

## 5 EXPERIMENTS

### 5.1 EXPERIMENTAL SETTING

**Datasets.** We curate subsets from three public benchmarks Something–Something-v2 (SSv2) Goyal et al. (2017), BridgeV2 Walke et al. (2023), and Fractal Brohan et al. (2022) to ensure sufficient temporal coverage. We annotate a total of 20,000 human hand interaction videos and 40,000 gripper interaction videos for training. We organize all samples into structured manipulation templates and randomly sample two videos from the same template, one as the reference and the other as the target. We collect 240 samples for evaluation, Both reference videos and corresponding target videos of evaluation pairs are unseen during training. Considering that some comparative methods Zhang et al. (2025); Zhao et al. (2024) require additional training for different reference videos, we further split the evaluation samples into 48 groups, where each group contains one reference video and five target images to ensure uniform coverage across different manipulation classes. Detailed data curation is provided in the Appendix A.

**Comparison Methods** We compare our method with representative image-to-video motion transfer approaches: (1) DynamiCrafter Xing et al. (2024) and CogVideoX Yang et al. (2024), further adapted via one-shot fine-tuning on the reference video; (2) MotionClone Ling et al. (2024), a training-free motion cloning framework; (3) MotionDirector Zhao et al. (2024), a dual-path LoRA-based model that decouples appearance and motion learning; and (4) FlexiAct Zhang et al. (2025), a learnable global motion transfer method. Except for MotionClone, all baselines require extra fine-tuning with the reference video. Training details and hyperparameters are provided in Appendix B.

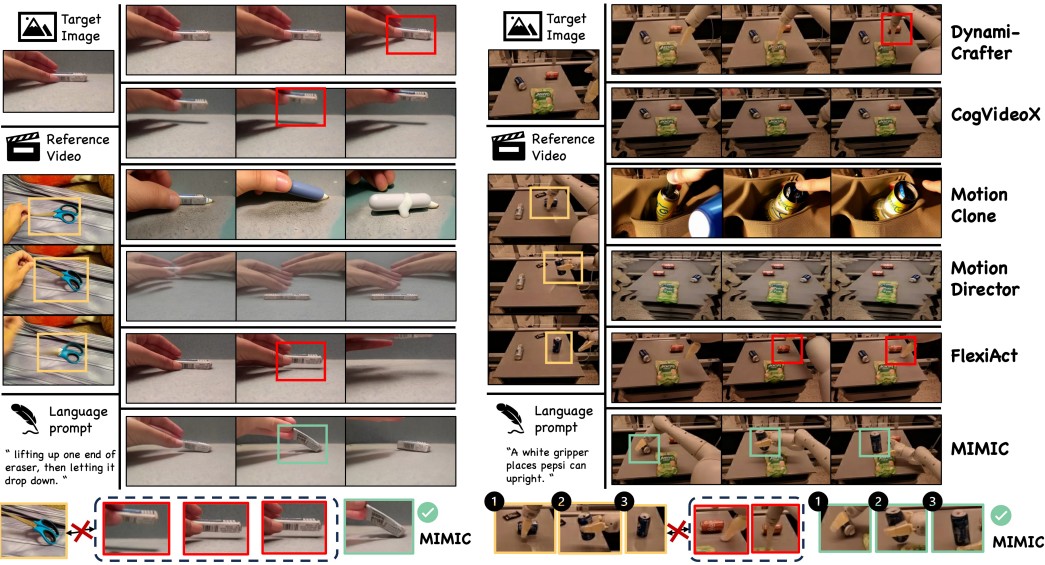

Figure 3: **Qualitative comparison of video motion transfer results.** Each row shows frames generated by a different method conditioned on the same reference video in two challenging scenarios. The **yellow box** marks the object state in the reference video; the **red box** marks the object state in the comparison video; and the **green box** marks the object state in our video.

**Implementation Details.** Our training dataset consists of 16-frame videos with a resolution of $320 \times 512$ pixels. Both stages are initialized from pretrained DynamiCrafter Xing et al. (2024) weights and optimized with AdamW on two NVIDIA H100 (80GB) GPUs. The batch size is set to 4 and the learning rate to $1 \times 10^{-5}$ throughout training. In **Stage I**, we freeze the temporal layers and finetune only the spatial and IMA layers of the UNet backbone. We first train for 5,000 steps on repeated first frames to preserve spatial fidelity, and then train for 50,000 steps on full video sequences to learn motion. In **Stage II**, we finetune the video generation model for 25,000 steps. The trainable components consist of the query encoder, the pair encoder, and the control module initialized from UNet weights. To stabilize training, we introduce the adaptive region loss after 5,000 steps and apply a sigmoid-based nonlinear warm-up over 2,000 steps to mitigate abrupt loss changes. At inference time, we adopt 50 DDIM sampling steps and set the CFG scale to 7.5.

## 5.2 EVALUATION

**Evaluation Metrics.** Our metrics consist of four aspects. Firstly, we employ CLIP-based Radford et al. (2021) text alignment and appearance consistency to reflect **Perceptual Similarity**. Secondly, **Temporal Quality** is evaluated using subject consistency and background stability from VBench Huang et al. (2024). Thirdly, we utilize the multimodal large language model (**MLLM**) Bai et al. (2023); Lin et al. (2023) in evaluation, as it possesses superior semantic understanding capabilities. Finally, we conduct extensive **Human Preference** evaluations via user studies. More details can be found in Appendix. C and C.3.

**Quantitative and Qualitative Analysis.** Our quantitative evaluation results are summarized in Tab. 1 and Fig. 3. Our method achieves the best performance in terms of temporal quality and appearance consistency, demonstrating its superior capability in preserving visual fidelity. It also performs strongly in text alignment, ranking second only to MotionClone. However, MotionClone underperforms on other metrics and exhibits weaker preservation of input image fidelity. Traditional quantitative metrics tend to emphasize low-level pixel alignment, but they are limited in their ability to assess aspects that require higher-level semantic understanding. For example, such metrics cannot reliably determine whether an object with a specific pose has been lifted correctly, or whether the model interacts precisely with the intended object in multi-object scenes. To address this limitation, we additionally incorporate evaluations using a multimodal large language model(MLLM) Lin et al. (2023), which possesses strong high-level semantic reasoning capabilities. Due to space constraints in Tab. 1, we report two dimensions that are most relevant to manipulation tasks, Interaction

Table 1: Quantitative comparison of manipulation video generation. We report several automatic evaluation metrics alongside human preference rates. Participants are asked to select the **top2** videos, making the subjective metric more robust and reliable.

| Method | Extra Finetune | Perceptual Similarity | | Temporal Quality | | MLLM Evaluation | | Human Preference |
| | | Text Alignment ↑ | Appearance Consistency ↑ | Subject Consistency ↑ | Background Stability ↑ | Interaction Rationality ↑ | Semantic Similarity ↑ | |
|---|---|---|---|---|---|---|---|---|
| DynamiCrafter | ✓ | 0.2684 | 0.8784 | 0.9185 | 0.9331 | 3.0543 | 2.4348 | 8.86% |
| CogVideoX | ✓ | 0.2667 | 0.8537 | 0.8128 | 0.9200 | 3.1318 | 2.3736 | 18.78% |
| MotionClone | ✗ | **0.2947** | 0.7400 | 0.6833 | 0.8569 | 3.0957 | 2.1277 | 0.90% |
| MotionDirector | ✓ | 0.2658 | 0.8336 | 0.8542 | 0.9160 | 3.1489 | 2.4149 | 0.96% |
| FlexiAct | ✓ | 0.2694 | 0.8999 | 0.8921 | 0.9220 | 3.5529 | 2.5238 | 27.8% |
| One-Stage | ✗ | 0.2688 | 0.8709 | 0.8591 | 0.9130 | 3.6170 | 2.4468 | – |
| w/o IMA Attention | ✗ | 0.2548 | 0.8537 | 0.8418 | 0.9029 | 3.6216 | 2.4134 | – |
| w/o Pair Prompt Control | ✗ | 0.2677 | 0.8862 | 0.9172 | 0.9213 | 3.8789 | 2.7526 | – |
| **MIMIC** | ✗ | 0.2721 | **0.9084** | **0.9291** | **0.9385** | **4.1381** | **2.9127** | **42.88%** |

Rationality and Semantic Similarity, while additional dimensions are provided in Fig. 4. In the MLLM-based evaluation, our method demonstrates superior performance in terms of operational completeness, accuracy, and semantic consistency with the reference video. Finally, we conduct a human preference study. To mitigate extreme biases, participants are asked to select their **top2** preferred results out of 6 candidates. Our method consistently receives a clear majority of preferences, further validating its effectiveness in generating manipulation videos that align with human judgment. Details on the MLLM evaluation can be found in the Appendix. C.2.

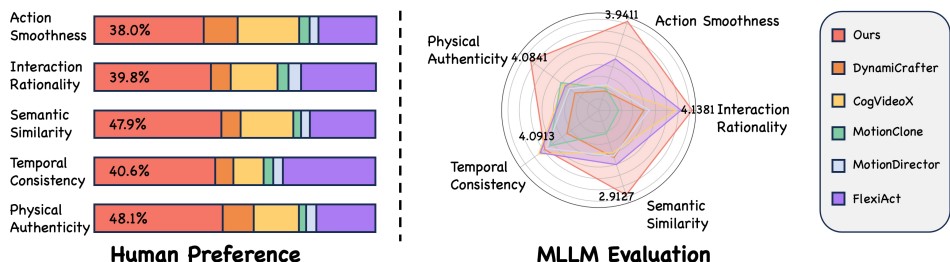

Figure 4: **Qualitative results of Human Preference and MLLM Evaluation. Right**: human preference selection ratios, where participants are asked to choose the **top2** videos from all options from various perspectives. **Left:** MLLM scores for the generated videos across different evaluation dimensions.

## 5.3 ABLATION STUDY

**Two-Stage vs. One-Stage** We first validate the rationale behind the proposed two-stage generation strategy, which progresses from motion patterns to visual details, by conducting an ablation study on single-stage direct video generation. In this experiment, all training settings remain consistent with those used in our Stage I model training, the ground truth video frames are used as supervision signals instead of masks. As shown in Fig. 5, videos generated by the diffusion model in a single stage suffer from severe visual quality issues. However, we observe that the interaction motion information conveyed roughly matches the reference videos and our results. This further confirms the rationality of using Stage I to learn interaction motion patterns by generating masks.

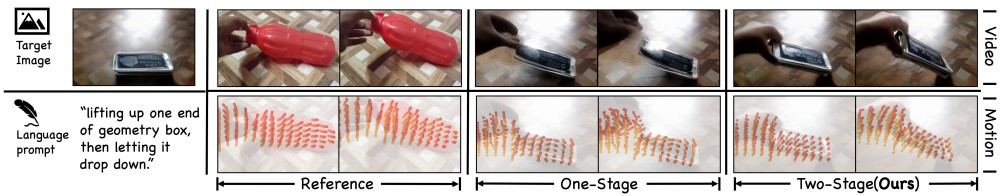

Figure 5: The results of one-stage and two-stage(ours) generation strategies.

**Effect of IMA and Pair Prompt Control** To verify the effectiveness of our approach in generating semantically consistent videos from reference videos, we conduct ablation studies on the **IMA**

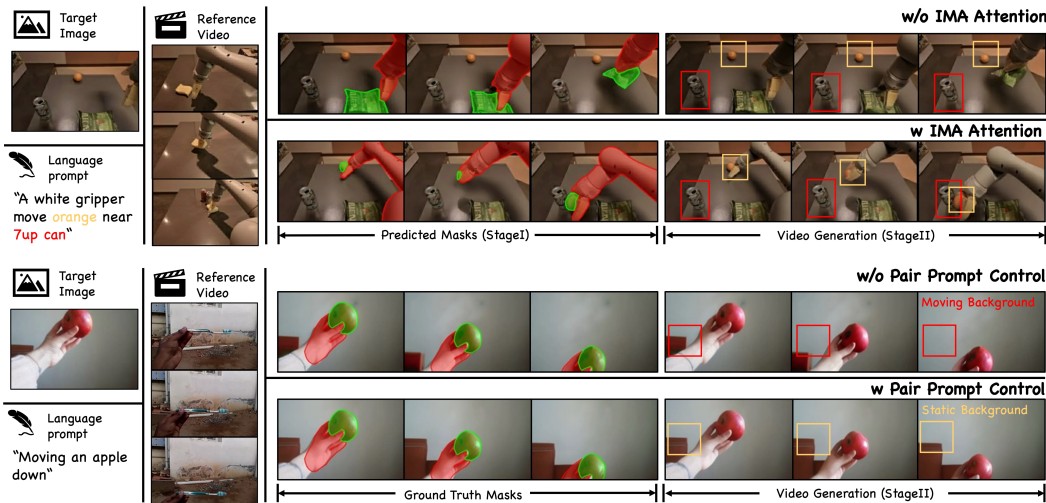

Figure 6: **Qualitative results of ablation study.** We show the interaction mask from Stage I and the generated video from Stage II. On the left, IMA attention enables semantic motion alignment with the reference video, while without it the video fails to match the language prompt. Right: Pair Prompt Control improves mask understanding; without it, the background moves with the mask, resembling camera motion rather than interaction. With Pair Prompt Control, the background remains stable, and changes arise only from true motion.

**Attention** module in Stage I and the **Pair Prompt Control** module in Stage II. Quantitative results are presented in Tab. 1. It can be observed that removing the IMA Attention module (w/o IMA) leads to a clear drop in **Semantic Similarity** (2.41 vs. 2.91) and degrades other metrics due to the lower-quality masks predicted in Stage I. As shown in the top of Fig. 6, the model misinterprets the prompt and manipulates the wrong object, illustrating the semantic inconsistency caused by the absence of IMA. Without Pair Prompt Control, the explicit mask struggles to capture complete manipulation information due to the coupling of object motion and camera movement, which leads to a slight performance decline across various metrics. As shown in the example (Fig. 6, bottom), the background drifts with the mask, indicating that the model generates camera motion instead of the hand–object interaction. Overall, IMA preserves high-quality semantic mask prediction, whereas Pair Prompt Control disentangles camera motion and injects fine-grained appearance information. Additionally, we provide further ablation studies on the region loss used in Stage II, detailed in the Appendix. D.

## 6 CONCLUSION

In this paper, we present MIMIC, a two-stage image-to-video generation framework designed for manipulation scenarios, which effectively leverages semantic information extraction and explicit interaction masks to produce physically plausible and semantically consistent videos. Our approach overcomes key challenges of existing methods by disentangling camera and object motions through a Pair Prompt Control mechanism, and enhancing temporal stability with an adaptive region loss. **Limitations and future work**: Limited by the capacity of the base model Xing et al. (2024), videos generated by our method are currently restricted to a maximum of 16 frames, which precludes the generation of long-horizon videos depicting complex operations. Adopting more powerful foundational models can facilitate our approach to synthesizing longer temporally coherent videos with more complex action compositions.

## 7 ACKNOWLEDGMENT

This work was supported in part by NSFC under No.62233013, 62293511, Key Research and Development Program of Zhejiang Province under No. 2025C01064, Fundamental and Interdisciplinary Disciplines Breakthrough Plan of the Ministry of Education of China under No.

JYB2025XDXM103. We extend our sincere gratitude to Qingtao Liu and Haoming Li for their insightful discussions and writing assistance.

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

## A DATA CURATION

### A.1 DATA PROCESS

Due to limitations of the pre-trained model, we only use videos with a length of 16 frames for training, which generally requires temporal sampling of the original data. However, this often leads to issues such as excessively large frame-to-frame variations and difficulty in capturing the full extent of the language prompt within the video, both of which significantly impact our training process. Additionally, motion-blurred frames frequently occur in video data, especially in the Something-Something-v2 (SSv2) Goyal et al. (2017) dataset. Such blurring poses challenges for our training process, as it can affect the consistency and accuracy of learned motion representations. We have processed the data as follows:

(1) *Redundant Frame Elimination*: We utilize OpenCV to detect the magnitude of motion both of the initial frames and the final frames of each video segment to determine whether the scene remains static. Frames exhibiting negligible motion at either end are identified as stationary and subsequently removed to eliminate redundant static frames.

(2) *Blurred Frame Elimination*: We utilize OpenCV to implement a method that calculates the variance of the Laplacian of each frame, providing an indicator of its blur level. When this indicator falls below a preset threshold, the frame is identified as blurry and consequently discarded.

(3) *Temporal Sampling*: After removing redundant and blurry frames, we perform uniform sampling over the entire video and set a maximum allowable sampling interval. When the interval between sampled frames exceeds this threshold, indicating excessive variation between frames, the corresponding segment is discarded.

## A.2 DATA ANNOTATION

Given a raw video $V$ paired with a language prompt $c$, we generate the annotations following the stream below:

(1) *Objects Segmentation*: We employ Grounded SAM2 Ren et al. to achieve open-set segmentation of interactive objects, categorizing target objects into two types: active manipulators (e.g., hands, grippers) and passive manipulated objects. For segmenting active manipulators, we predefine comprehensive language prompts according to the data set used. For example, 'human hand' in SSv2 Goyal et al. (2017) and 'white robotic gripper' in Fractal Brohan et al. (2022). For passive object segmentation, 'placeholder' language labels have been provided in SSv2. For the Fractal Brohan et al. (2022) and BridgeV2 Walke et al. (2023) datasets, we extract the first noun from the prompt $c$ as input to Grounded SAM2.

(2) *Manipulation Template*: We adopt a classification approach consistent with SSv2 Goyal et al. (2017), which categorizes videos using templates such as "move [something] down." Specifically, we implement a natural language processing pipeline with spaCy to extract structured action-object pairs from textual descriptions. The pipeline first normalizes the input text and then identifies the main verb in each sentence, including phrasal verbs like "place [something] upright" and "pick up [something]", enabling precise action extraction aligned with established dataset standards. The structured language is used solely for this categorization step; at inference time, the model only requires a natural-language description and a reference video, without relying on structured prompts.

## A.3 DISSUSION ABOUT MOTION REPRESENTATION

We adopt a mask-based motion representation to model human–object interactions. This choice is motivated by the need for reliable supervision: high-quality global optical flow Teed & Deng (2020), point trajectories Karaev et al. (2024), or part-level optical flow obtained by combining Co-TrackerKaraev et al. (2024) with CMP Zhan et al. (2019) as used in MOFA-Video Niu et al. (2024) are difficult to obtain for large-scale videos, particularly when objects undergo non-rigid or highly deformable motion. As shown in Fig. 7, these representations all degrade significantly in cloth-folding scenarios, where global flow, sparse tracking points, and part-level flow fail to capture coherent motion cues. In contrast, segmentation masks either manually annotated or automatically extracted via SAM2 Ravi et al. (2024) remain stable and consistent even under such challenging conditions.

Moreover, masks inherently encode the semantic separation between the manipulated object and the hand or gripper, a property essential for role-conditioned motion generation in our pipeline. Alternative motion representations lack this role-aware structure and are more susceptible to failures under large motions or complex deformations. While mask-based signals may limit the modeling of fine-grained shape dynamics, they provide interpretable and robust motion cues well suited to our task.

## B DETAILS ON BASELINES

*DynamiCrafter* Xing et al. (2024). We fine-tune the spatial layers of DynamiCrafter for 5,000 steps on ground truth videos and captions from the specific dataset to mitigate the existing domain gap. And we further fine-tune on the reference video for an additional 300 steps. During inference, we use 50 DDIM sampling steps and set the guidance rescale to 0.0.

*CogVideoX* Yang et al. (2024). Unlike our method, which uses 16-frame videos for training, to preserve the prior knowledge of the pre-trained model, we fine-tune CogVideoX using 49-frame videos. Similar to DynamiCrafter, we employ LoRA to fine-tune CogVideoX on the reference video for 5,000 steps to bridge the domain gap, and we further train on the reference video for an additional

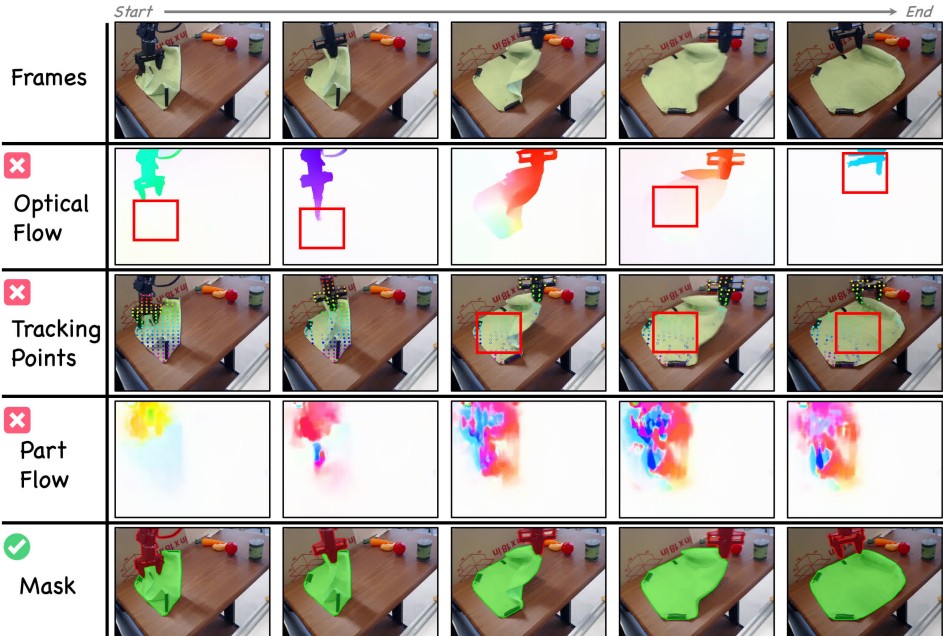

Figure 7: **Motion representation comparison in a cloth-folding scenario.** Global optical flow (Row 2) deteriorates rapidly under large non-rigid deformation, leading to unstable or missing estimates in critical regions. Tracking points (Row 3) fail to maintain consistent correspondences on the deforming cloth, resulting in drifting or jittery trajectories that do not capture coherent motion. Part-level flow (Row 4) suffers from even greater instability, as part assignments fluctuate across frames and produce noisy, visually inconsistent flow fields. In contrast, segmentation masks (Row 5) remain temporally stable and accurately preserve the global structure of the cloth throughout the folding sequence.

300 steps for one-shot scenarios. During inference, we set the LoRA rank to 128 and use 50 DDIM sampling steps.

*MotionClone* Ling et al. (2024). MotionClone is a training-free method, which extracts motion priors from the temporal attention matrix of a reference video and constructs an energy function to guide the sampling process. During inference, we set the number of inference steps to 100 and the number of guidance steps to 40.

*MotionDirector* Zhao et al. (2024). We follow the setup of MotionDirector by first training the spatial LoRA on the target image for 300 steps, followed by training the temporal LoRA on the 16-frame reference video for 150 steps. During inference, we set the noise prior to 0 and use DDIM with 50 sampling steps.

*FlexiAct* Zhang et al. (2025). We follow the FlexiAct recommendation by training the frequency-aware embedding(FAE) on the 49-frame reference video for 3,000 steps. During inference, we set the transition timestep for the FAE to 0.8, the additional attention weight in the FAE to 1.0, and the guidance scale to 6.0.

## C EVALUATION

### C.1 AUTOMATIC METRICS

We provide a detailed explanation of the principles and meanings behind the automatic evaluation metrics used in our experiments:

(1) **Perceptual Similarity:** The metric includes *Text Alignment* as the average cosine similarity between the text prompt embedding and frame embeddings encoded by CLIP Radford et al. (2021),

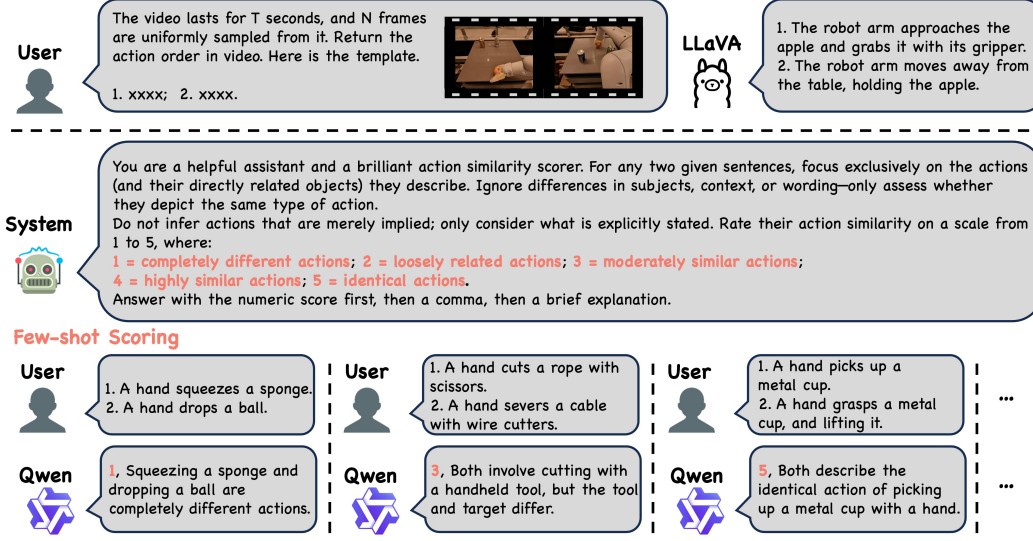

Figure 8: Detailed pipeline for computing semantic similarity based on large-scale pretrained models.

and *Appearance Consistency* as the average visual similarity between the first frame and subsequent frames to capture intra-video coherence.

(2) **Temporal Quality** Huang et al. (2024): A human-aligned spatiotemporal benchmark assessing *Subject Consistency* and *Background Stability*. For Subject Consistency, it measures whether the appearance of a subject remains consistent throughout the video by calculating DINO Oquab et al. (2023) feature similarity across frames. For Background Stability, it evaluates the temporal consistency of background scenes by calculating CLIP Radford et al. (2021) feature similarity across frames.

## C.2 Details of MLLM Evaluation

**Semantic Similarity** To establish a reliable metric for evaluating the semantic consistency between the reference video and the generated output, we utilize the prior knowledge embedded in large-scale pretrained models. Specifically, we temporally concatenate the reference and generated videos, inserting a buffer of two blank frames between the two segments to construct a combined video $V_{concat}$. Leveraging the video understanding capability of LLaVA-Video-7B Lin et al. (2023), we generate two captions that respectively describe the content of each video segment within $V_{concat}$.

Next, to assess the similarity between these captions at a fine-grained semantic level, we employ Qwen2.5-7B-Instruction Bai et al. (2023), a large language model with advanced language comprehension. This model evaluates the action similarity based solely on the explicitly stated actions—ignoring differences in subjects, context, or wording—assigning a score on a 1 to 5 scale, where 1 denotes completely different actions and 5 indicates identical actions.

The scoring process starts by presenting the model with pairs of captions and instructing it to rate the similarity based exclusively on the described actions. The model is asked to first provide a numeric score, followed by a concise explanation. Fig. 8 illustrates the detailed pipeline of this semantic similarity computation, including some of the example prompts used to guide the models and the standardized scoring templates designed for Qwen.

**Other MLLM Metrics** For the other four aspects, we only use LLaVA-Video-7B Lin et al. (2023), which has video understanding capabilities. To improve the reasoning behind the MLLM scoring, we guide it with multiple related questions before requesting a score and provide clear criteria corresponding to each score. We show the prompt input for Interaction Rationality as an example in Fig. 9.

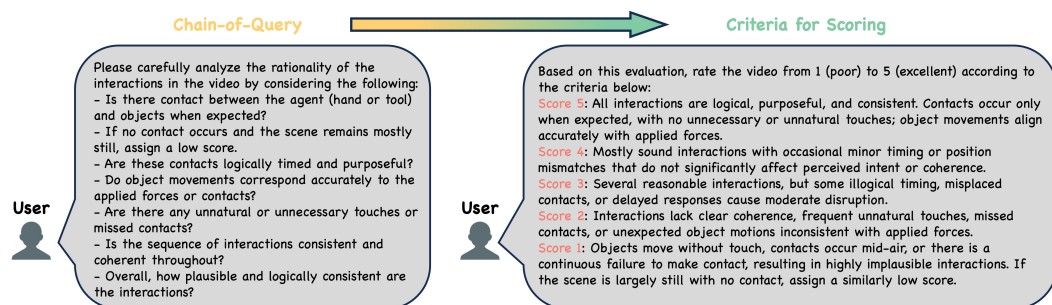

Figure 9: **Example of MLLM Evaluation.** Taking Interaction Rationality as an example, we present the specific content used when employing the MLLM to evaluate manipulation videos.

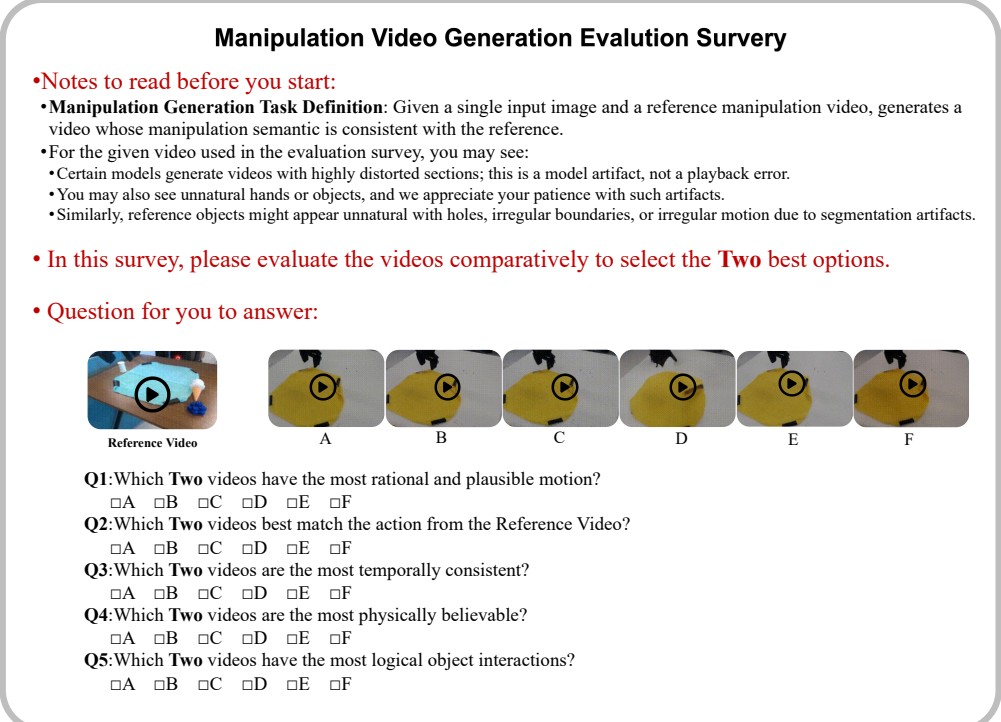

Figure 10: Human evaluation for video generation. We provide a reference video and an target image, users are asked to evaluate and select their preference. videos based on various video editing criteria.

## C.3 DETAILS OF HUMAN EVALUATION

We design a user study by randomly selecting 25 groups from our generated video library. Each group consists of a target image, a reference video, and videos generated by eight different methods, including ours. We invite 25 participants to perform a human evaluation. They are asked to select their two preferred videos among the eight generated ones based on three criteria: *Action Smoothness*, *Interaction Rationality*, *Temporal Consistency*, *Physical Authenticity*, and *Semantic Consistency* with the reference video. As shown in the Fig. 10, we present the content of the user study along with the specific question settings. The Tab. 2 presents the detailed results of the human evaluation. Our method demonstrates a clear advantage in the human preference study, outperforming competing approaches across all evaluated criteria.

Table 2: Quantitative comparison on human preference evaluation.

| Method | Action Smoothness | Ineraction Rationality | Physical Authenticity | Temporal Consistency | Semantic Similarity |
|---|---|---|---|---|---|
| DynamiCrafter | 13.8% | 6.9% | 8.4% | 7.7% | 6.6% |
| CogVideoX | 22.4% | 22.1% | 18.4% | 11.8% | 19.2% |
| MotionClone | 0.4% | 0.7% | 0.5% | 2.4% | 0.5% |
| MotionDirector | 0.5% | 0.9% | 1.2% | 0.5% | 1.7% |
| FlexiAct | 24.9% | 29.6% | 23.4% | 37.0% | 24.1% |
| **MIMIC** | **38.0%** | **39.8%** | **48.1%** | **40.6%** | **47.9%** |

Table 3: Quantitative comparison on MLLM evaluation.

| Method | Action Smoothness | Ineraction Rationality | Physical Authenticity | Temporal Consistency | Semantic Similarity |
|---|---|---|---|---|---|
| DynamiCrafter | 3.2173 | 3.0543 | 3.5217 | 3.8804 | 2.4348 |
| CogVideoX | 3.6923 | 3.1318 | 3.6681 | **4.1468** | 2.3736 |
| MotionClone | 2.8404 | 3.0957 | 3.7021 | 4.0425 | 2.1277 |
| MotionDirector | 3.2978 | 3.1489 | 3.5957 | 4.0744 | 2.4149 |
| FlexiAct | 3.7761 | 3.5529 | 3.6441 | 4.1323 | 2.5238 |
| **MIMIC** | **3.9411** | **4.1381** | **4.0841** | 4.0913 | **2.9127** |

# D  ADDITIONAL ABLATION STUDY

## D.1  ADAPTIVE REGION LOSS

In Stage II, we design an **Adaptive Region Loss** to address consistency issues and introduce a hyperparameter $\lambda$ as shown in Equation. 4. We conduct additional ablation experiments to validate the effect of this loss. To reduce the influence of other factors, we use ground truth masks as inputs and quantitatively analyze the impact of the region loss using metrics that evaluate visual quality, specifically *Appearance Consistency*, *Subject Consistency*, and *Background Stability*. Due to computational resource limitations, the ablation experiments requiring retraining for the Adaptive Region Loss are conducted on a subset of approximately 10,000 videos from the SSv2 Goyal et al. (2017) dataset, rather than on our full-scale dataset. And we only evaluate on human hand interaction data. We first validate the composition and specific effect of the Adaptive Region Loss, which,

Table 4: Ablation study on the different compositions and weights of Adaptive Region Loss.

| $\lambda$ | Appearance Consistency ↑ | Subject Consistency ↑ | Background Stability ↑ |
|---|---|---|---|
| ori. | 0.8927 | 0.8894 | 0.9296 |
| w/o FM | 0.9042 | 0.9118 | 0.9360 |
| 0.25 | 0.9001 | 0.8975 | 0.9328 |
| 0.50 | 0.8872 | 0.8709 | 0.9225 |
| 0.75 | 0.9043 | 0.9039 | 0.9325 |
| 1.00 | 0.9054 | 0.9090 | 0.9323 |
| 1.25 | 0.9033 | 0.8937 | 0.9339 |
| 1.50 | 0.8972 | 0.9118 | 0.9326 |

as formulated, consists of two mask components: the overall motion mask and the temporally repeated first-frame mask. We design three experimental groups for evaluation: the original diffusion loss(ori.), the Adaptive Region Loss without the first-frame mask(w/o FM), and the complete Adaptive Region Loss(full). Furthermore, we perform hyperparameter ablation for $\lambda$ in the range of 0 to 1.5 with increments of 0.25. The detailed experimental results are presented in the accompanying Tab. 4.

As illustrated in Fig. 11, we present visualization results under different compositions and weights of the loss function. When the mask for the first frame is omitted, noticeable ghosting artifacts appear. Conversely, as the weight increases, the visual details contained within the mask improve significantly.

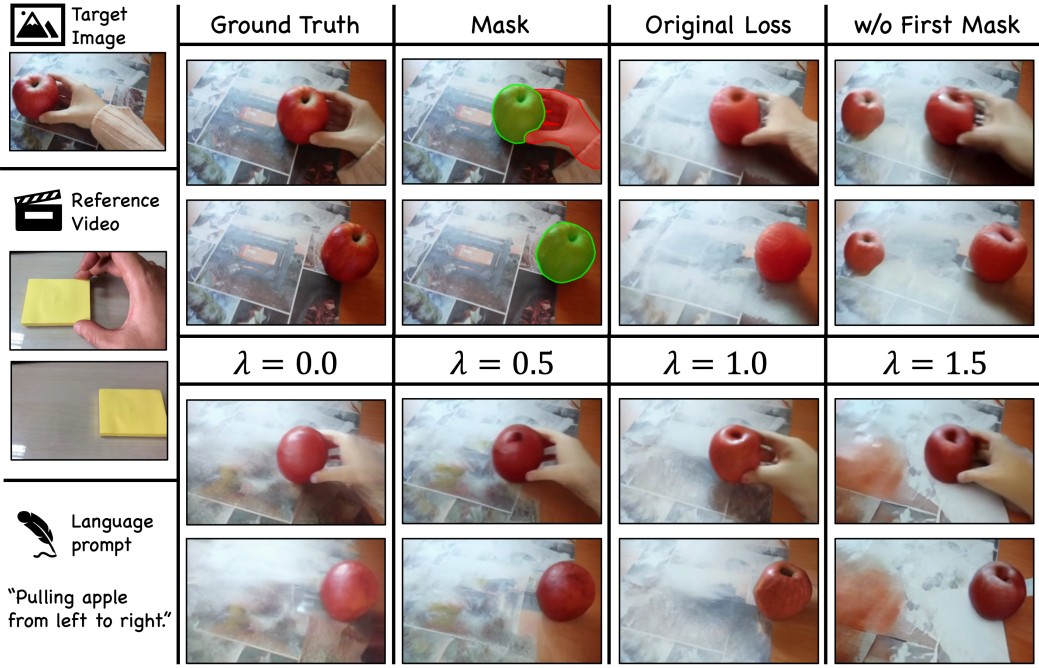

Figure 11: The generation results under different compositions and weights of the loss function.

## D.2    DIFFERENT INPUT TYPE

we conduct an ablation study comparing two types of conditioning inputs for the motion generation model: (i) interaction masks only, and (ii) the full input image. The results are reported in Tab. 5. We observe that using mask-only input leads to significantly worse performance than using the image. This degradation stems from the fact that masks provide only coarse spatial localization and lack essential appearance cues required for understanding the underlying scene context. Without access to the image content, the model tends to overly rely on the temporal changes of the masks themselves and fails to infer how the object should be manipulated in a physically plausible manner.

Table 5: Ablation study of different input type.

| Input Type | Perceptual Similarity | | Temporal Quality | | MLLM Evaluation | |
|---|---|---|---|---|---|---|
| | Text Alignment ↑ | Appearance Consistency ↑ | Subject Consistency ↑ | Background Stability ↑ | Interaction Rationality ↑ | Semantic Similarity ↑ |
| **Mask** | 0.2672 | 0.8804 | 0.8745 | 0.9168 | 3.5689 | 2.5862 |
| **Image** | 0.2721 | 0.9084 | 0.9291 | 0.9385 | 4.1381 | 2.9127 |

As illustrated in Fig. 12, models conditioned solely on masks often attempt to replicate the motion pattern observed in the reference video (e.g., unfolding a piece of cloth) rather than generating a contextually appropriate manipulation trajectory for the target image. This indicates that mask-only conditioning encourages the model to focus on mask dynamics rather than action semantics or scene geometry, leading to incorrect or unrealistic motion synthesis. In contrast, using the full image provides rich appearance information, enabling the model to better understand object attributes, hand–object configurations, and scene layout. This contextual grounding is essential for generating a plausible motion trajectory rather than merely propagating mask deformations. These findings justify the necessity of Stage I: although models like Grounding-SAM2 Ren et al. can directly provide interaction masks, conditioning on the full image allows the generative model to build its own understanding of the scene, leading to more accurate and contextually appropriate motion generation.

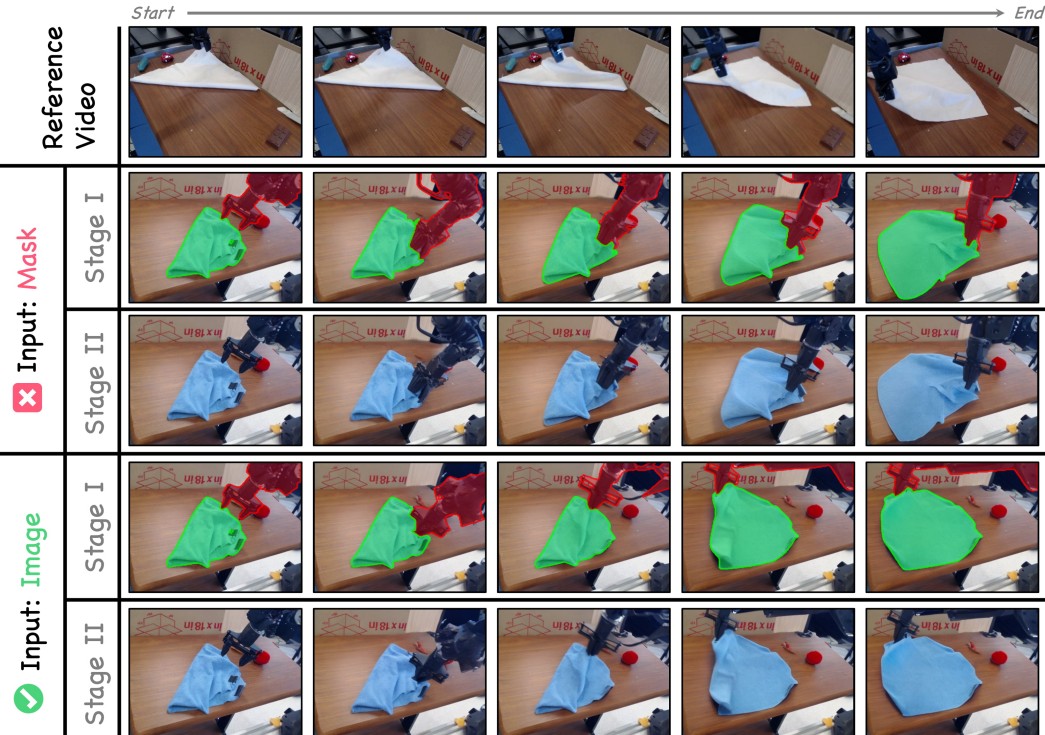

Figure 12: Motion representation comparison in a cloth-folding scenario.

# E NETWORK ARCHITECTURE

## E.1 QUERY ENCODER AND PAIR ENCODER

In Stage II, we design two encoders, $E_{tar}$ and $E_{ref}$, to separately capture the motion information of the target video and the semantic information of the reference pair. Both $E_{tar}$ and $E_{ref}$ have a similar lightweight network architecture, as illustrated in Fig. 13. The architecture mainly consists of convolutional layers interleaved with SiLU activation layers. Through convolutional downsampling operations, the input conditional information is aligned to the size of the latent code $z_t$. Notably, in the last convolutional layer, we employ the zero convolution layer from ControlNet Zhang et al. (2023). This zero convolution layer is initialized with zero weights and biases, ensuring that the initial output of the model remains identical to the output of the pretrained model.

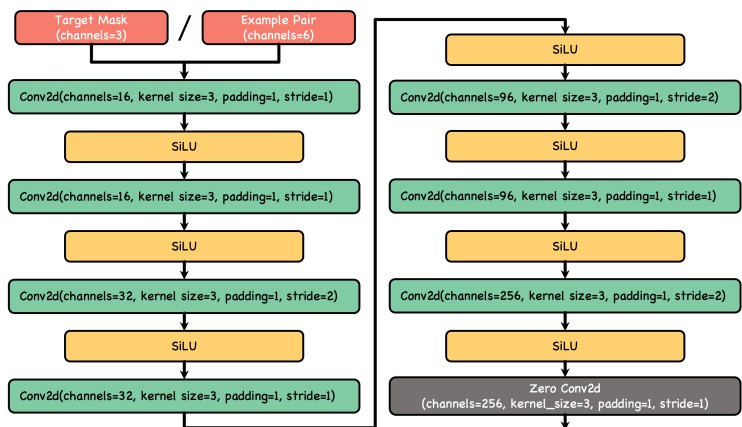

Figure 13: Network Architecture of Query Encoder and Pair Encoder.

## F ADDITIONAL INTERACTION SCENARIOS

To further examine the flexibility of the proposed framework beyond the settings covered in the main experiments, we evaluate its behavior under a broader range of interaction scenarios involving changes in embodiment, coordination structure, and temporal horizon.

**Cross-Domain Transfer.** We examine whether motion cues extracted from human-hand demonstrations can be transferred to robotic end-effectors. As illustrated in Fig. 14, the model is able to interpret high-level manipulation intent from human-hand videos and generate plausible interaction sequences for a parallel-jaw gripper. Despite the substantial differences in appearance and morphology, the generated motions maintain consistent contact patterns and task-relevant dynamics, indicating that the representation can accommodate changes in manipulator embodiment.

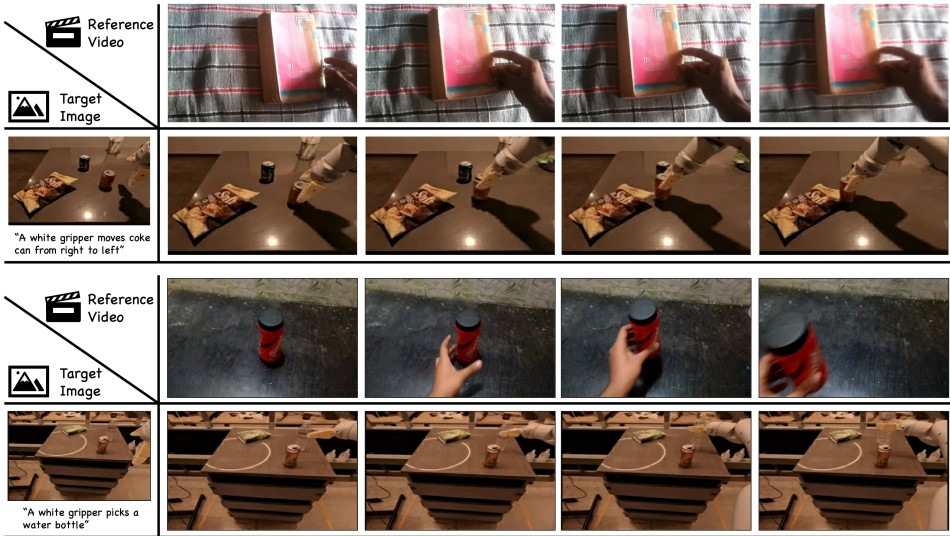

Figure 14: Cross-domain generation from human-hand demonstrations to a robotic gripper. The model preserves task-relevant contact patterns and motion intent across embodiments.

**Two-Hand Coordination.** We analyze the performance of our method in coordinated two-hand interactions. As illustrated in Fig. 15, the model generates coherent predictions for scenarios involving two human hands jointly manipulating an object. The outputs preserve stable spatial relations between the hands and exhibit coordinated motion trajectories, suggesting that the MIMIC remains effective even when the interaction involves multiple effectors operating in a shared workspace.

**Long-Horizon Generation.** We further examine whether longer videos can be produced by sequentially composing multiple generations. As shown in Fig. 16, we generate a long-horizon interaction sequence by using the final frame of a generated segment as the initial frame for the next generation. This process is repeated across three segments guided by different reference videos. The resulting long video maintains smooth transitions between segments, preserves consistent object and manipulator configurations, and exhibits coherent temporal evolution throughout the sequence. These observations indicate that the proposed framework can be extended to produce temporally longer interaction videos through iterative composition.

## G MORE RESULTS

We present additional comparisons with other methods, including interaction videos between human hands or mechanical grippers and objects, as shown in Fig. 18–21. We showcase additional generated videos of human hands and mechanical grippers interacting with objects in Fig. 22 and 23. MIMIC not only produces videos that semantically match the reference video but also distinguishes

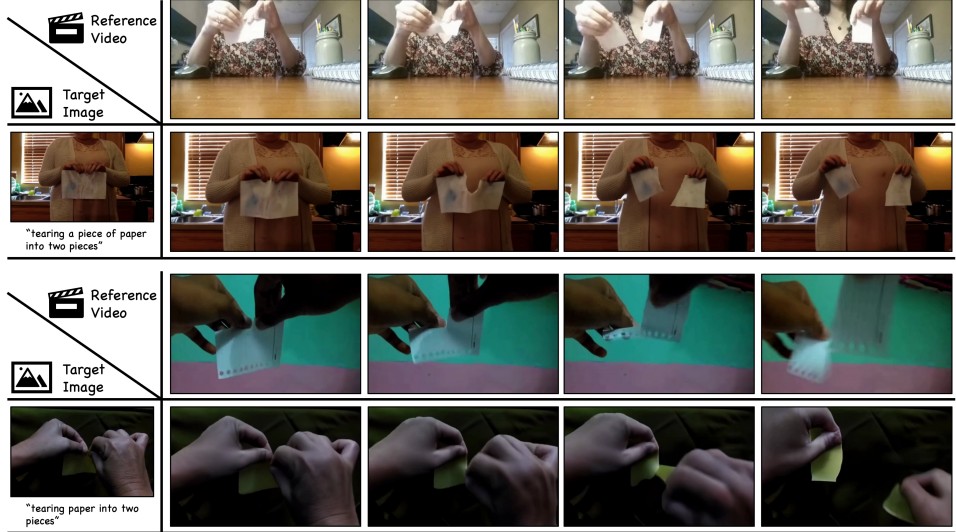

Figure 15: Generation results for coordinated two-hand manipulation, showing stable spatial relations and synchronized motion between the two hands.

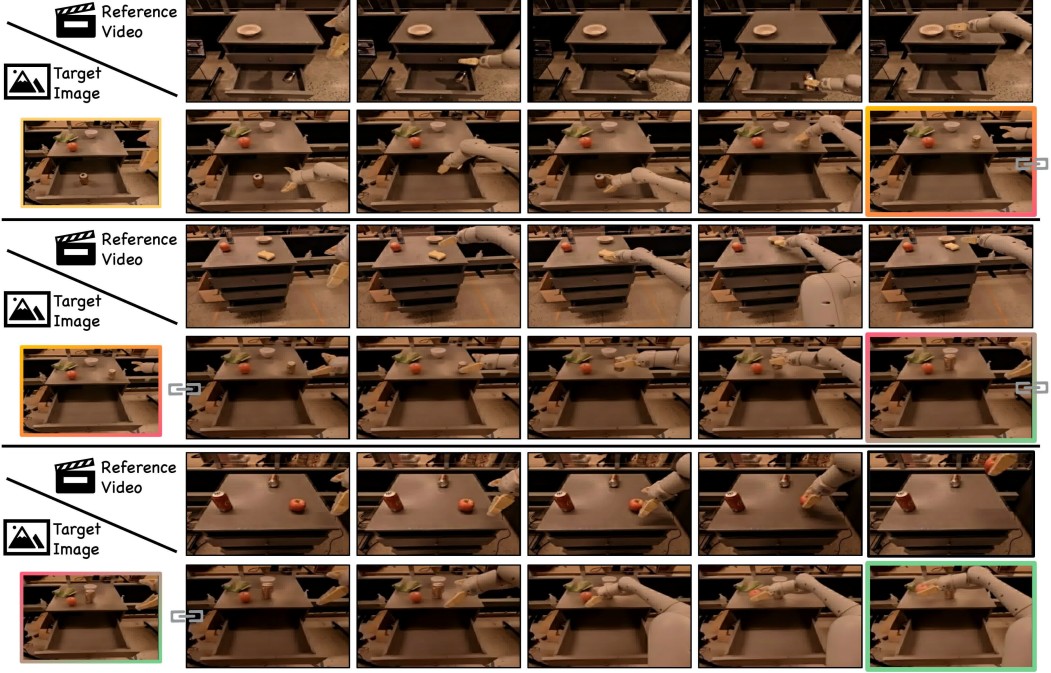

Figure 16: Long-horizon video obtained by iteratively composing segments, using final frame of each segment as initial frame of the next segment to ensure smooth transitions.

between similar operations, such as pulling and pushing. Moreover, our method is capable of handling interactions with non-rigid objects, such as unfolding.

## H FAILURE CASES

While our method achieves strong results across various interaction scenarios, it still struggles in some challenging cases, as illustrated in Fig. 17. The upper portion of the figure shows a case

with **significant occlusion**, where large parts of the hand and object are not visible. This leads to noticeable appearance instability and incorrect hand–object scale in the first predicted frame, which then propagates through subsequent frames and degrades the overall generation quality. The lower portion of the figure depicts an interaction involving **fine-grained dexterous manipulation**, specifically twisting open a bottle cap. This task requires precise rotational control, stable contact maintenance, and detailed reasoning about fingertip–object interactions. Our model is not yet able to fully capture these subtle motion dynamics, resulting in imperfect motion synthesis.

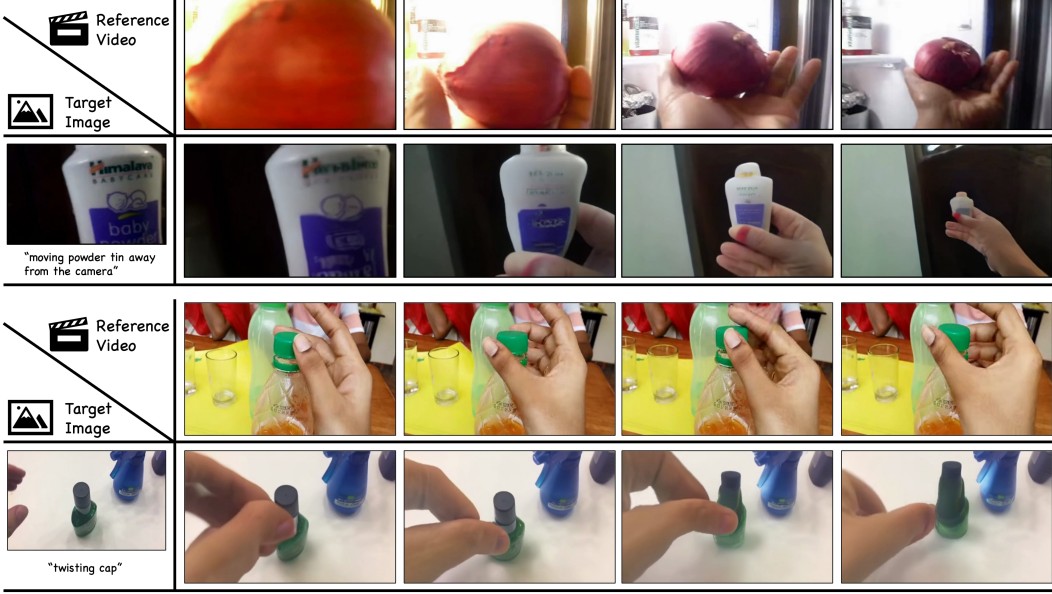

Figure 17: **Failure cases.** Top: severe occlusion leads to incorrect hand–object scale and unstable generation. Bottom: fine-grained dexterous manipulation such as twisting a bottle cap remain challenging, resulting in inaccurate motion synthesis.

## I    BROADER IMPACT

This project addresses the critical bottleneck of scarce and costly real interaction data in the field of embodied intelligence by generating a large volume of plausible hand or gripper manipulation videos. This effectively facilitates research and applications in robotic perception and manipulation, accelerating technology dissemination and cost reduction. Meanwhile, the diversity of generated data enhances the generalization capabilities of robotic systems and fosters interdisciplinary innovation. By providing rich, low-cost data resources, the approach opens new technical pathways and practical possibilities for the development of embodied intelligence, yielding significant social and industrial impacts.

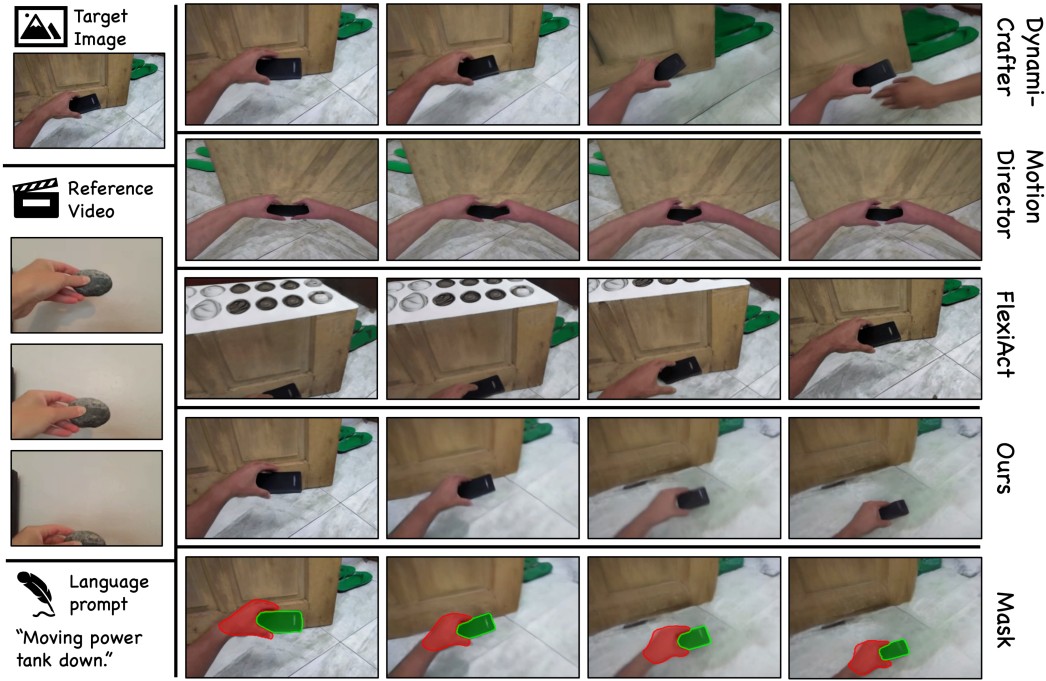

Figure 18: More comparison results of human hand-object interaction video from the reference video.

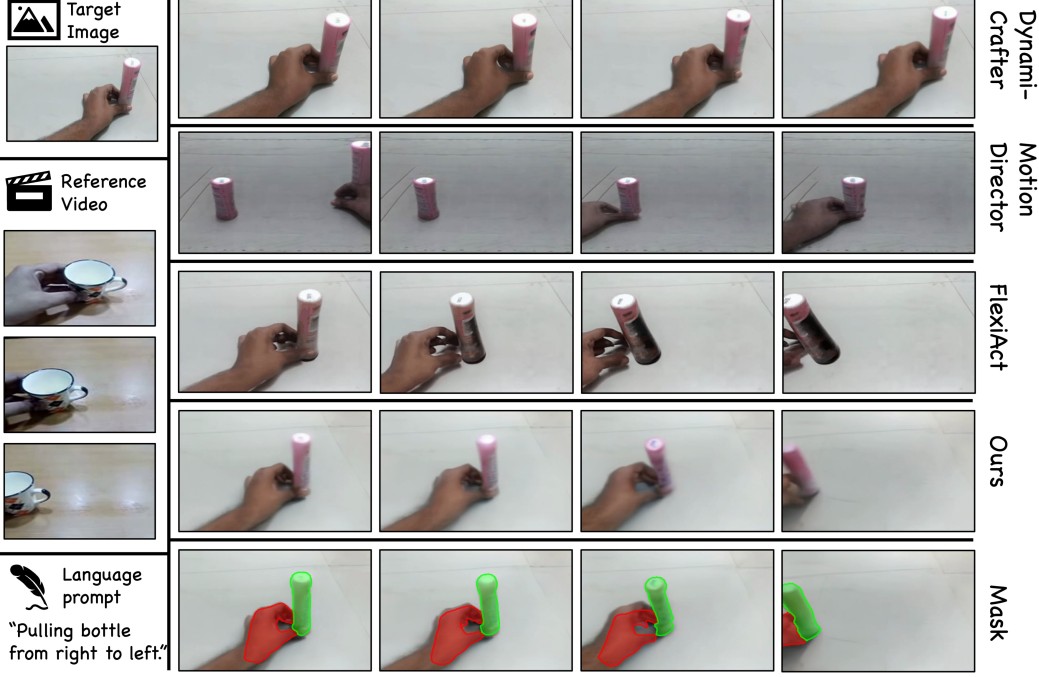

Figure 19: More comparison results of human hand-object interaction video from the reference video.

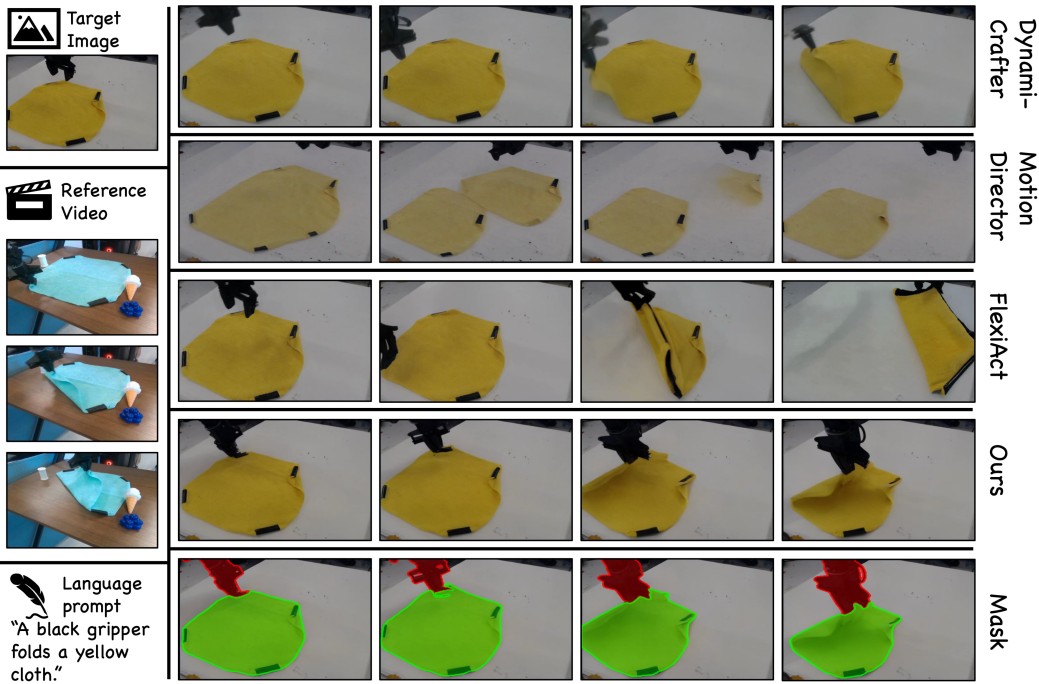

Figure 20: More comparison results of robotic gripper-object interaction video from the reference video.

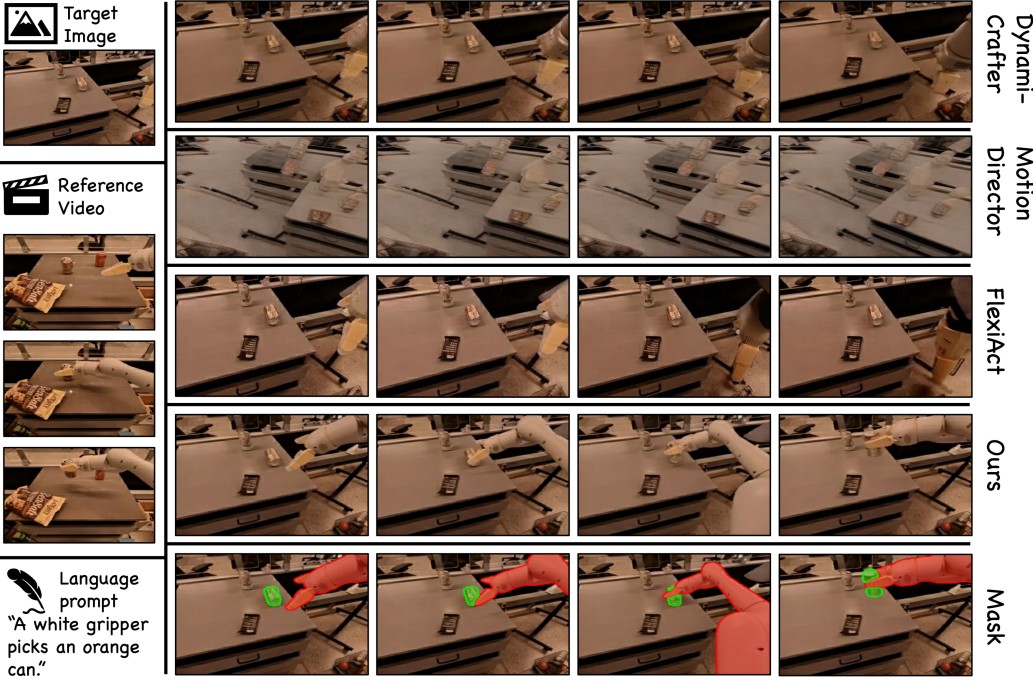

Figure 21: More comparison results of robotic gripper-object interaction video from the reference video.

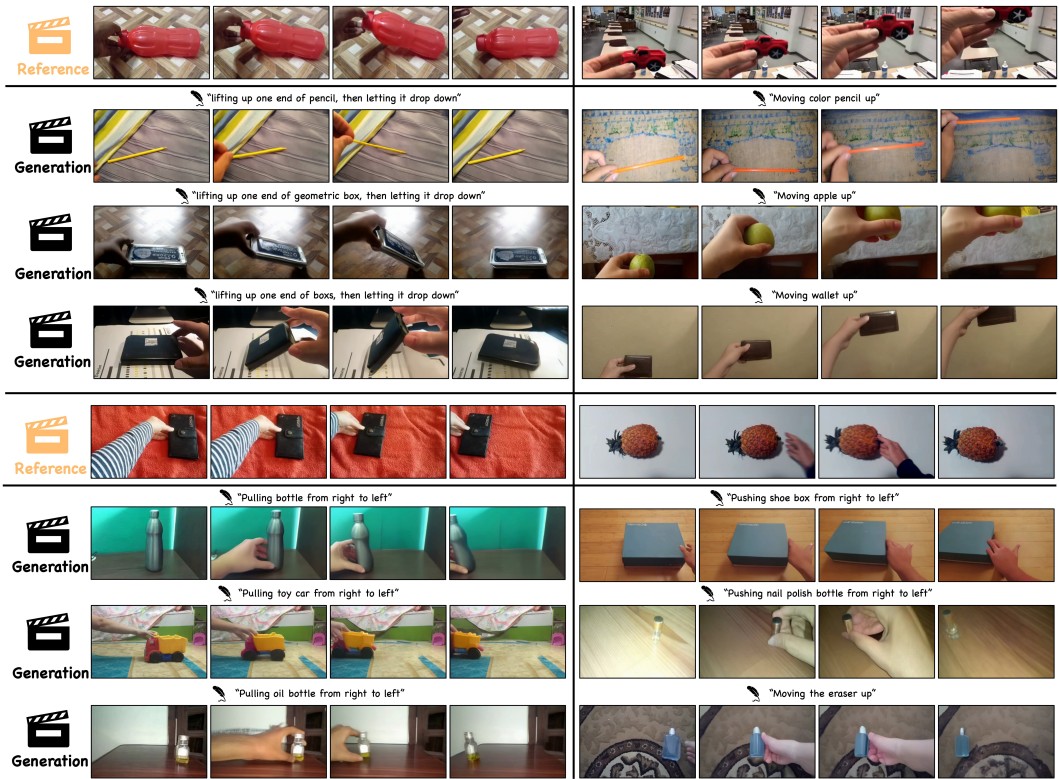

Figure 22: Results of human hand-object interaction video from reference videos.

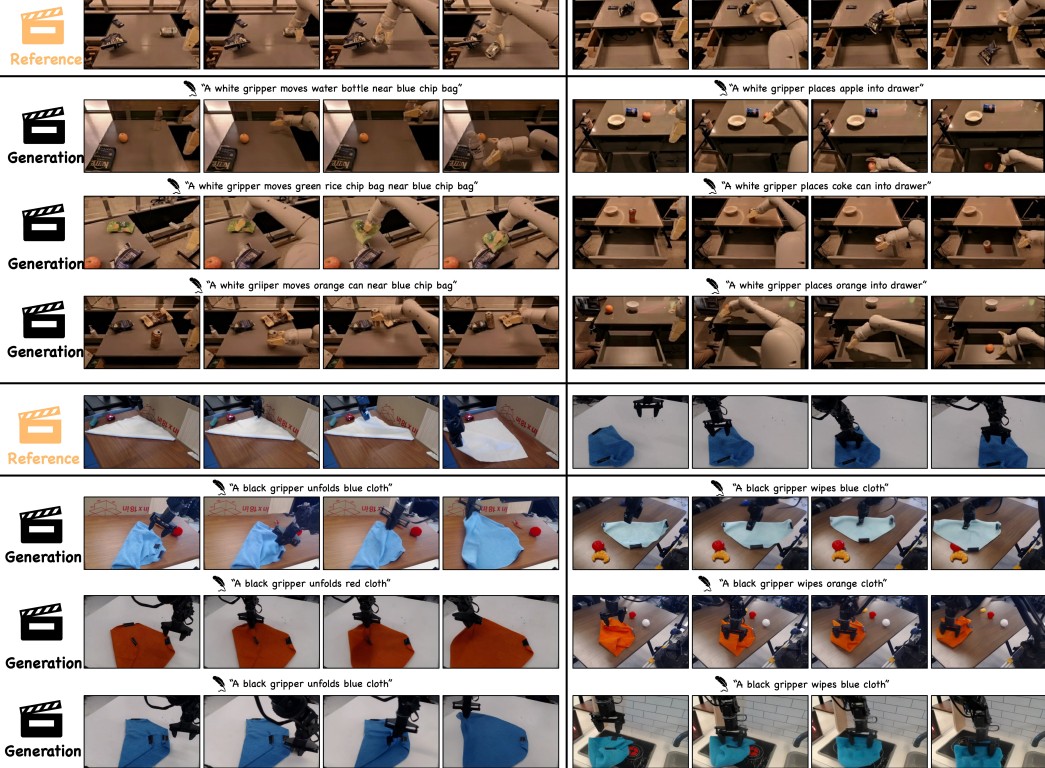

Figure 23: Results of robotic gripper-object interaction video from reference videos.

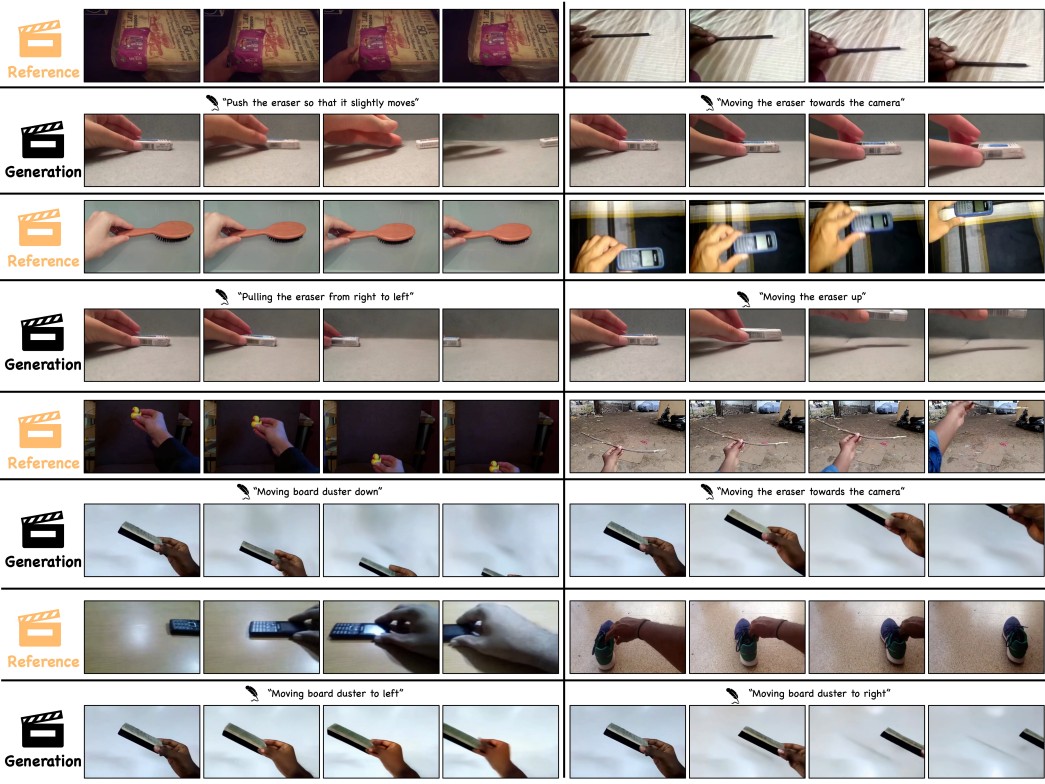

Figure 24: Results of human hand video on same target image with different reference videos.

