# OpenReview forum: "MIMIC: Mask-Injected Manipulation Video Generation with Interaction Control"
_ICLR.cc/2026/Conference — ICLR 2026 Poster_

### Official Review · Reviewer_Sdzw · 2025-10-29

**Soundness:** 4
**Presentation:** 4
**Contribution:** 4
**Rating:** 8
**Confidence:** 3

**Summary:**

This paper proposes a two-stage, image-to-video diffusion framework for generating manipulation videos with fine control over object interactions. First, an Interaction-Motion-Aware (IMA) module extracts semantic masks from a reference video to guide future frames. Then, these masks are injected into a diffusion model to steer the video generation process. A Pair Prompt Control mechanism helps disentangle object motion from camera motion by conditioning on both the target image and the reference video.
The method is shown to preserve manipulation intent and detail, even when dealing with deformable or diverse objects, outperforming existing baselines in both visual fidelity and control.

**Strengths:**

1. The paper addresses an interesting and important problem with a novel solution.
2. The paper proposes several new ideas that could be leveraged for future research in different direction, including the IMA attention, Pair Prompt Control module and the idea of using ICL for learning motion from motion masks of similar tasks.
3. Experiments are sound and the visual results show the superiority of MIMIC over previous work.

**Weaknesses:**

1. The method depends on relevant reference videos. This might limit its applicability in scenarios where such references are not available or are very different from the target.
2. It is unclear how the reference videos are chosen.
3. The method is limited to generating 16-frame videos.

**Questions:**

1. How did the authors choose the reference videos?
2. Did the authors try to generate longer videos by concatenating generations? (e.g. use the last frame of a previously generated video as the initial frame of a new generation)
3. Did the authors try multi-object or multi-hand interaction scenarios? e.g. manipulate different objects with 2 hands or a hand and a gripper in the same video?
4. Are there failure cases you exhibited, such as specific objects/motions/lighting conditions etc.?

**Details Of Ethics Concerns:**

No concerns.

---

> ### Author Response · Authors · 2025-11-21
> **Response to Reviewer Sdzw**
>
> We thank the reviewer for the positive and encouraging feedback, we address the remaining concerns and questions below.
>
> **Q1: The method might limit its applicability in scenarios where such references are not available or are very different from the target.**
>
> **A1:** We appreciate the reviewer’s concern about the reliance on relevant reference videos. However, our method is motivated by the assumption that users can provide semantically related reference videos, which we believe is a practical setting. Our framework is designed to leverage such semantically similar references to enhance generation quality, consistency, and controllability.
>
> Providing reference videos very different from target is analogy to providing text prompts different from the scene, which may leads the model confusing. Besides, without such references, the task would essentially revert to generic video generation, which serves as a baseline of our method.
>
> **Q2: How did the authors choose the reference videos?**
>
> **A2:** Thank you for pointing this out. In the data preparation stage, we first extract the *manipulation template* for each video, which specifies the manipulation type and its associated parameters. To obtain these templates, we apply a simple NLP pipeline that extracts verb–noun structures to determine the manipulation type. Based on these templates, we group all videos into corresponding categories. When selecting reference videos, we randomly sample from the set of videos that share the same manipulation template as the target video. This ensures that the reference videos are consistent and comparable in terms of manipulation settings.
>
> The detailed procedure for defining and annotating manipulation templates is provided in Appendix A.2, please kindly check out.
>
> **Q3: Did the authors try to generate longer videos by concatenating generations?**
>
> **A3:** We appreciate the reviewer’s insightful suggestion.As suggested, we used the last frame of a generated segment as the initial frame for the next segment and repeated this process with three different reference videos. This allowed us to produce a longer, continuous video composed of multiple reference-driven segments. Detailed results can be found in the Appendix F, please kindly refer to Fig.16.
>
> **Q4: Did the authors try multi-object or multi-hand interaction scenarios?**
>
> **A4:** Thank you for the thoughtful question. In Appendix F(Fig.15), we provide additional results where our method show effectiveness in synthesis of collaborate multi-hand ineraction, such as tearing a piece of paper. Please kindly check it out.
>
> **Q5: Are there failure cases you exhibited, such as specific objects/motions/lighting conditions etc.?**
>
> **A5:** We appreciate the reviewer’s suggestion to include failure cases. We have added representative examples to the Appendix H(Fig.17). As shown in the upper example, the model struggles under **significant occlusion**, where missing visual cues lead to unstable appearance changes and incorrect hand–object scale that propagates across the generated frames. The lower example illustrates a case of **fine-grained dexterous manipulation**, specifically twisting open a bottle cap, where the required precise rotational motion and continuous contact reasoning remain challenging for the model to capture, leading to inaccurate motion synthesis. These examples highlight current limitations in handling heavy occlusions and highly intricate interaction dynamics, suggesting important directions for improving robustness in future work.

---

> > ### Comment · Area_Chair_xz4x · 2025-11-26
> >
> > Dear reviewer Sdzw:
> >
> > Could you take a look at the author's response and leave your feedback.
> >
> > AC

---

### Official Review · Reviewer_Kzh4 · 2025-10-30

**Soundness:** 3
**Presentation:** 2
**Contribution:** 3
**Rating:** 4
**Confidence:** 4

**Summary:**

This paper proposes MIMIC, a two-stage diffusion framework to generate manipulation videos by leveraging a reference video for semantic and motion cues. The method first generates a sequence of interaction masks (Stage I) and then renders the final video (Stage II), aiming to provide a scalable data source for embodied AI. The authors demonstrate strong quantitative and qualitative performance over existing methods.

**Strengths:**

- The paper addresses the important and challenging problem of data scarcity for training embodied intelligence systems.
- The proposed two-stage framework is logical. Decoupling the generation process into (1) motion/interaction understanding (mask generation) and (2) high-fidelity rendering (video generation) is a sensible approach to decompose a complex problem.
- The qualitative comparisons provided in the Supp. video are compelling.
- The ablation studies are informative and effectively validate the contributions of the proposed components, i.e., the IMA Attention and the Pair Prompt Control mechanisms .

**Weaknesses:**

- Relation to Video Editing: The related work section focuses on "Video Motion Customization" and "Interaction Video Generation" but does not explicitly discuss the field of generative video editing. The task in this paper shares conceptual overlap with reference-based video editing. It would strengthen the paper to include this discussion, clarify the key distinctions, and situate the work relative to recent highly relevant papers in the HOI image/video editing domain, such as [1][2].
- The paper does not clearly explain how the (reference, target) video pairs are constructed for training. Is a video simply used as its own reference? Are pairs formed in some other way (based on action label)? This is a critical methodological detail that is currently missing and needs to be explicitly stated.
- The paper's description of Stage I is confusing. It's trained to "identify" the manipulated object , but Grounding-SAM2 is already used for annotation. Why train a generative model for a recognition task? Why Grounding-SAM2 isn't used at test time to provide the initial object mask, allowing the diffusion model to focus on its core task: generating the new motion pattern (i.e., the mask trajectory)?
- It is unclear what the scope of the tested actions and objects is. For instance, do the experiments demonstrate generalization to unseen object categories, or new instances of seen categories? The experimental setup and data splits should be described more explicitly to clarify what level of generalization is being evaluated.
- About the use of language: Appendix A.2 says that the method relies on structured language templates (e.g., "move [something] down") and an NLP pipeline to "extract structured action-object pairs". I feel this should be explicitly stated in the main paper, as it clarifies that the model is not conditioned on free-form, natural language prompts, right?
- About the masks: the text repeatedly describes them as "binary masks". However, numerous figures visualize multi-class masks, with hands/grippers and manipulated objects colored differently. The authors need to clarify whether the masks are binary (e.g., interaction vs. background) or multi-class (e.g., hand vs. object vs. background). Also Ln204 “soft binary mask”, what does “soft” mean?

[1] Affordance Diffusion: Synthesizing Hand-Object Interactions

[2] HOI-Swap: Swapping Objects in Videos with Hand-Object Interaction Awareness

**Questions:**

My overall feeling is that the paper presents a strong method, but the presentation is unclear in several critical areas, as detailed in the comments. I hope the authors can clarify these points in their response. Happy to increase my score once these are clarified.

---

> ### Author Response · Authors · 2025-11-21
> **Response to Reviewer Kzh4 (1/2)**
>
> We sincerely thank the reviewer for the thoughtful and encouraging feedback. We appreciate the recognition of the importance of our problem setting, the soundness of our two-stage framework, and the effectiveness of the IMA Attention and Pair Prompt Control modules. We address the reviewer’s remaining concerns in detail below.
>
> **Q1: Relation to video editing**
>
> **A1:** We appreciate the reviewer’s suggestion to discuss the connection to generative video editing. We have updated the Related Work section to include recent HOI-oriented video editing methods. While our task shares conceptual overlap with reference-based editing, the problem setting is fundamentally different. Video editing approaches such as Affordance Diffusion [1] and HOI-Swap [2] focus on **locally modifying existing content** while preserving the original motion, background, and interaction structure. Typical examples include inserting a hand into an object-only image or replacing the manipulated object in a hand–object interaction video. In contrast, our method tackles **interaction-centric Image-to-Video (I2V) generation**, where the model is required to synthesize a new hand–object interaction sequence from a single static image, rather than transforming an existing video. This distinction makes our work goes beyond the scope of conventional editing pipelines.
>
> **Q2: How the video pairs are constructed for training?**
>
> **A2:** We construct training pairs by first organizing all training samples into **structured manipulation templates(action label)**, which group videos that share the same action type. During training, we **randomly sample two videos from the same template**, one as the reference and the other as the target. This ensures that the model learns to transfer a motion pattern across different scenes but similar interaction semantics, which is essential for our in-context generation framework. We sincerely thank the reviewer for highlighting the need for clarification, and we have revised the manuscript accordingly to provide the above details.
>
> **Q3: Why Stage I is trained in a joint perception and generation manner?**
>
> **A3:** We thank the reviewer for the thoughtful question. We found that training Stage I in a joint perception and generation manner substantially enhances the model’s ability to understand scene context, including object geometry, hand–object configuration, and task-related spatial constraints. These visual cues are crucial for predicting physically consistent manipulation trajectories. The image inputs allow the diffusion model to implicitly recognize the manipulated object and the effector, and to reason about how they should interact within the scene capabilities that cannot be achieved with masks alone. In contrast, providing only an initial mask at test time removes most of the scene information. This limits the model to tracking mask deformations rather than understanding action semantics or physical affordances.
>
> We provide the corresponding ablation results in Appendix D.2. Both quantitative and qualitative results confirm that mask-only conditioning significantly degrades performance. As shown in Fig.12, mask-only models tend to imitate the motion pattern in the reference video (e.g., unfolding cloth) rather than producing a context-appropriate trajectory for the target scene. This indicates that mask inputs encourage the model to track mask deformations instead of reasoning about action semantics or scene geometry. To facilitate the reviewer’s examination, we additionally include the quantitative results here.
>
> | Input Type | Text Alignment | Appearance Consistency | Subject Consistency | Background  Consistency | Interaction Rationality | Semantic Similarity |
> | ---------- | -------------- | ---------------------- | ------------------- | ----------------------- | ----------------------- | ------------------- |
> | Mask       | 0.2672         | 0.8804                 | 0.8745              | 0.9168                  | 3.5689                  | 2.5862              |
> | Image      | 0.2721         | 0.9084                 | 0.9291              | 0.9385                  | 4.1381                  | 2.9127              |
>
>
> [1] Affordance Diffusion: Synthesizing Hand-Object Interactions (CVPR 2022)
>
> [2] HOI-Swap: Swapping Objects in Videos with Hand-Object Interaction Awareness (NeurIPS 2024)

---

> ### Author Response · Authors · 2025-11-21
> **Response to Reviewer Kzh4 (2/2)**
>
> **Q4: What the scope of the tested actions and objects is?**
>
> **A4:** All evaluation pairs both their reference videos and their corresponding target videos are *unseen* during training, after constructing all (reference, target) pairs offline, we randomly allocate 10% of the full set as a validation split. The evaluation set contains **(1) unseen object categories**, **(2) new instances of seen categories** with different shapes, colors, or textures, and **(3) unseen backgrounds and environments** that differ in layout, lighting, and clutter. This setup ensures that the reported results reflect the ability of model to generalize to unseen object–action combinations.
>
> **Q5: Use of structured language templates.**
>
> **A5:** The structured templates and the NLP pipeline are used **only during training** to categorize manipulation types and construct the reference–target pairs described above. Importantly, this does not imply that the model depends on structured prompts at inference time. During inference, the model only requires a natural-language description of the target video and a semantic-similar reference video; it does not require the structured templated form used during training. This clarification has been added to the Appendix A.2.
>
> **Q6: About the masks**
>
> **A6:** We apologize for the lack of clarity and the bug in writing. The reviewer is right. Currently we use three-class mask(hand/gripper vs. object vs. background). During dataset construction with Grounded-SAM2, we first obtain binary masks for the hand/gripper and the manipulated object. We then assign distinct RGB values to correspond region. Regarding the phrase “soft binary mask,” it is bug in writing. We sincerely thank the reviewer for pointing it out, we have updated the main manuscript accordingly.

---

### Official Review · Reviewer_xAXU · 2025-11-01

**Soundness:** 3
**Presentation:** 3
**Contribution:** 3
**Rating:** 4
**Confidence:** 4

**Summary:**

This paper proposes MIMIC, a novel two-stage image-to-video diffusion framework designed to generate realistic and controllable manipulation videos. The method first utilizes an Interaction-Motion-Aware (IMA) module to generate a sequence of semantic interaction masks guided by a reference video. Subsequently, it introduces a Pair Prompt Control mechanism that conditions the final video generation on both the predicted masks and the original reference video, effectively disentangling object motion from camera motion and enhancing controllability.

**Strengths:**

1. The authors' decomposition of the task into two stages, "mask generation" and "video rendering," is a reasonable design. Furthermore, the authors demonstrate the necessity of this decomposition through ablation studies (such as "Two-Stage vs. One-Stage" and Figure 6) and clearly show the contributions of each key module (IMA and PPC).

2. The authors identified the problem with using only masks for control. The Pair Prompt Control (PPC) part is a clever solution. Figure 6 (background drift vs. static background) shows how this part is effective at separating object motion from camera motion.

**Weaknesses:**

1. The IMA module just copies the motion from the reference video and does not understand the physics of the target scene. The predicted masks in Figure 6 support this, appearing to be a direct motion transfer that ignores the specific geometry and constraints of the target object.

2. The examples in the paper are all very similar, showing only robot-to-robot or hand-to-hand tasks. This makes it unclear if the method can work for more general cases where the reference and target are different. For instance, it is not shown if a human hand video can be used to control a robot arm.

3. The method's first stage requires reference manipulation masks as input, but the paper does not explain how these are obtained. Class-agnostic models like SAM are difficult to automatically and accurately prompt to segment only the manipulated object in complex videos. Meanwhile, class-specific semantic segmentation models (e.g., Mask R-CNN) would restrict the method to only predefined object categories, causing it to fail on any new objects.

**Questions:**

1. The experiments only show in-domain cases. Have the authors tested cross-domain generalization? For example, can a video of a human hand be used to guide a robot arm?

2. Figure 6 shows that lacking the IMA module leads to failure. With the IMA module, how well does it work if the reference motion is complex or the object is occluded?

3. The paper mentions that the input of Stage I includes text, but it appears to be unused in the architecture (Figure 2) and the IMA module description. Could the authors clarify how exactly the text is used in Stage I?

---

> ### Author Response · Authors · 2025-11-21
> **Response to Reviewer xAXU**
>
> We sincerely thank the reviewer for the detailed and constructive feedback, as well as for recognizing the value and contributions of our two-stage formulation and the demonstrated effectiveness of the IMA and PPC modules.
>
> **Q1: The IMA module simply copies reference motion?**
>
> **A1:** No. With respect, there may exist a misinterpretation of Figure 6. In the reference video, the robot arm moves **from the bottom-left toward the top-right**. In the model with IMA (bottom row), the generated motion goes **from the top-right toward the bottom-left**, which is different from the reference video. This shows that the model is adapting to the target object and scene constraints showing the capability of scene-aware and physics-aware motion synthesis. In contrast, the model without IMA (top row) produces motion **from the bottom-left toward the top-right** indicating that the model collapses into copying the reference rather than reacting to the prompt-specified target object.
>
> **Q2: Cross-domain generalization (e.g., human→robot):**
>
> **A2:** We appreciate the reviewer’s suggestion. To demonstrate the effectiveness of our method in cross-domain motion transfer, we additionally provide corresponding experiments results in Appendix F (Fig. 14), for example, by using a human-hand reference video to guide the synthesis of robot-arm motion. The results demonstrate that our framework can still produce semantically consistent motion patterns under such domain shifts.
>
> **Q3: How reference manipulation masks are obtained:**
>
> **A3:** As we have briefly introduced in the Footnote of Page 5, the reference mask is obtained by the open-vocabulary segmentation model Grounding-SAM2[1] with corresponding language prompt. Specifically, to obtain the reference masks used in Stage I, we first extract the *manipulated object name* from each dataset’s textual label,  we then apply Grounded-SAM2 [1] to all videos offline to generate high-quality manipulation masks prior to training. In our experiments, the reference mask for both training and evaluation is precomputed for convenience.
>
> **Q4: On the challenge cases like the reference motion is complex or the object is occluded**
>
> **A4:** As shown in our main results(Fig.3) and additional examples(Fig.18-23), the proposed IMA module enables the model to handle a wide range of motion patterns, such as folding cloth, placing can upright. Moreover, we have added new experiments in Appendix F(Fig.15), including coordinated two-hand manipulation results. We believe these cases validate the effectiveness of our method in generating reasonably complex interaction dynamics. Please kindly check out.
>
> At the same time, we acknowledge that the model still encounters difficulties in **extremely challenging scenarios**. As detailed in Appendix H, heavy occlusion can obscure key hand–object configurations, leading to  inconsistent appearance or scale that accumulate across frames. Similarly, interactions that involve fine-grained dexterous manipulation, such as "twisting cap" manipulation, demand continuous contact reasoning that our methods cannot yet fully capture.
>
> **Q5: On the use of text input in Stage I**
> **A5:** Thank you for your thoughtful question. The text input is integrated into the diffusion model following the standard approach in Latent Diffusion Models (LDM) [2]. Specifically, the text embeddings are injected into the spatial layers via cross-attention, conditioning the denoising process on the structured action–object description.
>
> [1] Grounded SAM: Assembling Open-World Models for Diverse Visual Tasks (ArXiv)
>
> [2] High-Resolution Image Synthesis with Latent Diffusion Models (CVPR 2022)

---

> > ### Comment · Area_Chair_xz4x · 2025-11-26
> >
> > Dear reviewer xAXU:
> >
> > Could you take a look at the author's response and leave your feedback.
> >
> > AC

---

### Official Review · Reviewer_uEB3 · 2025-11-03

**Soundness:** 3
**Presentation:** 3
**Contribution:** 3
**Rating:** 6
**Confidence:** 3

**Summary:**

The paper proposes MIMIC, a two-stage image-to-video diffusion framework MIMIC designed for manipulation video generation. The propsoed training framework consists of two-stages: 1) Condition on a reference video, reference video interation mask, and target image, generte the target video interation mask. 2) condition on reference video, reference video interation mask, target image and masked target image, generate the target video. The paper use DynamiCrafter as the base model. The model was trained and evaluated on human-hand and robotic-gripper datasets built from SSv2, BridgeV2, and Fractal datasets. It was compared against major baselines such as DynamiCrafter, CogVideoX, MotionClone, FlexiAct, MotionDirector, etc.

**Strengths:**

- The paper is well-written and easy to follow, presenting its ideas and methodology clearly.

- The paper combines quantitative metrics, MLLM-based semantic evaluation, and human preference studies, providing a comprehensive assessment of model performance.

- The paper conducts extensive ablation studies, including analyses of the Two-Stage vs. One-Stage framework and the effectiveness of each key component, further validating the proposed approach.

**Weaknesses:**

- **Design choice of motion representation**: I was curious about the motivation for adopting masks as the abstraction for motion information rather than alternative representations such as tracking points or optical flow [1]. While masks provide clear spatial localization, they might be too rigid in preserving the object shape from the reference video, potentially limiting flexibility in motion adaptation.

Reference: [1] VideoJAM: Joint Appearance-Motion Representations for Enhanced Motion Generation in Video Models (ICML).

- **Impact of base model**: Would employing a more powerful base model (e.g., a larger or more recent video diffusion backbone) further improve the overall generation quality and temporal coherence? Since the paper mentions limitations due to the capacity of the current base model, it would be interesting to discuss how the proposed approach might scale with stronger foundational architectures.

**Questions:**

Please see the weaknesses.

---

> ### Author Response · Authors · 2025-11-21
> **Response to Reviewer uEB3**
>
> We thank the reviewer for the constructive feedback and we are encouraged by the positive comments regarding the clarity of the comprehensiveness of our evaluation protocol, the strength of our ablation studies, and the presentation, which together affirm the effectiveness of our proposed approach.
>
> **Q1: On the choice of motion representation**
>
> **A1:** Our choice of a mask-based motion representation is motivated by the fact that current mask-prediction models(SAM2[1]) offer **more robust and reliable** performance than alternatives such as global optical flow(RAFT[2]), sparse tracking points(CoTracker[3]), or part-level flow (CoTracker[2] + CMP[4]). This makes mask annotations substantially easier to obtain at scale. We have added Appendix A.3(Fig.7) to further illustrate this advantage, these alternative signals degrade severely in challenging scenarios such as cloth folding, where global flow becomes noisy, tracked points drift or disappear, and part-level flow fails to capture coherent motion. In contrast, segmentation masks remain consistent, robust, and semantically aligned across a wide range of interactions. Moreover, by encoding the masks with different colors to **indicate semantic roles** (green representing the manipulated object, red representing the hand or gripper), we inject explicit semantic separation that enables Stage II to produce more controllable and semantically aligned motion. In comparison, direct optical flow methods do not inherently encode such class-level semantics, making them less suitable for our goal of explicit, role-aware motion control.
>
> Nevertheless, we agree that with the development of perceptual models, high-quality motion representations will become increasingly accessible, enabling more effective guidance.
>
> **Q2: Would employing a more powerful base model further improve the overall generation quality and temporal coherence?**
>
> **A2:** Yes, we believe that a more powerful video diffusion backbone would further improve the generation quality and temporal coherence. Our method is built upon DynamiCrafter[5], which was the SOTA open-source I2V base model available when we initiated this project. We believe the core design of our framework is general and can be transferred to more powerful diffusion backbones, for example, the recently released Wan 2.2[6] (July 2025). To achieve this, one may still adopt our two-stage generation pipeline; recent Wan-based work such as AnimateAnything [7] has demonstrated the feasibility of such staged designs. For our in-context learning mechanism using a reference video, this can be integrated into a DiT-based backbone by injecting reference-conditioned features through fine-tuned extra cross-attention layers during the denoising process. Similar conditioning strategies have been validated in image editing tasks, such as RelationAdapter[8].
>
> This extension is in our plan, but the engineering effort and computational requirements exceed the time constraints of the rebuttal period. We will release the implementation once the migration is ready.
>
>
> [1] SAM 2: Segment Anything in Images and Videos (ICLR 2025)
>
> [2] RAFT: Recurrent All Pairs Field Transforms for Optical Flow (ECCV 2020)
>
> [3] CoTracker: It is Better to Track Together (ECCV 2024)
>
> [4] Self-Supervised Learning via Conditional Motion Propagation (CVPR 2019)
>
> [5] DynamiCrafter: Animating Open-domain Images with Video Diffusion Priors (ECCV2024)
>
> [6] Wan: Open and Advanced Large-Scale Video Generative Models (ArXiv)
>
> [7] AnimateAnything: Consistent and Controllable Animation for video generation (CVPR 2025)
>
> [8] RelationAdapter: Learning and Transferring Visual Relation with Diffusion Transformers (ArXiv)

---

> > ### Comment · Area_Chair_xz4x · 2025-11-26
> >
> > Dear reviewer uEB3:
> >
> > Could you take a look at the author's response and leave your feedback.
> >
> > AC

---

### Author Response · Authors · 2025-12-02
**Response Letter: Summary of Revisions and Updates**

Thank you for taking the time to review our work. Based on your valuable feedback, we have made several modifications and re-uploaded the updated PDF. The changes are summarized as follows:

- **For Reviewer uEB3:**
  - We add **discussion of motion presentation** in Appendix A.3.

- **For Reviewer xAXU:**
  - We add **cross-domain (hand-to-gripper) experiments** in Appendix F (Fig.14) to show that our method still generates semantically aligned robot motions, demonstrating robust cross-domain performance.
  - We include two **failure cases** under extremely challenging scenarios (such as heavy occlusion) in Appendix H.

- **For Reviewer Kzh4:**
  - We add **discussions of video editing** in the related work section.
  - We provide more details about the construction of **training and evaluation data** in the experimental section.
  - We add **a new ablation study** in Appendix D.2 to **analyze the impact of different input types (image vs. mask)** on generation quality, and both qualitative and quantitative results demonstrate the advantages of our chosen design.
  - We provide **a clearer explanation of the use of structural language templates** in Appendix A.2.
  - We revise "soft binary mask" to "mask".

- **For Reviewer Sdzw:**
  - We add additional experimental results on both **two-hand coordination** and **long-horizon generation** in Appendix F (Fig.15, Fig.16).
  - We add **failure cases** under several extremely challenging scenarios in Appendix H.

We believe that the above revisions, together with the detailed responses provided in the rebuttal, comprehensively address all concerns raised by the reviewers. We sincerely appreciate the reviewers’ constructive comments, which have helped us further strengthen the clarity, rigor, and completeness of the manuscript. Thank you again for your thoughtful feedback and for the opportunity to improve our work.

---

### Author Response · Authors · 2025-12-02
**General response**

We thank all reviewers for their thoughtful feedback and valuable insights.

We are encouraged that all reviewers recognize our work as “**addressing the important and challenging problem of data scarcity for training embodied intelligence systems**” (R. Kzh4) and as “**a strong method**” (R. Kzh4) with “**an interesting and important novel solution**” (R. Sdzw), “**could be leveraged for future research in different direction**” (R. Sdzw). Additionally, we are pleased that our two key designs: (a) the two-stage formulation is regarded as “**reasonable**” (R. xAXU) and “**logical**” (R. Kzh4), and (b) the Pair Prompt Control mechanism is highlighted as “**a clever solution**” that “**effectively disentangles object motion from camera motion**” (R. xAXU).

We further appreciate the reviewers’ positive reception of our experimental design. Our experiments are noted to provide “**a comprehensive assessment of model performance**” (R. uEB3). And the ablation studies are regarded as “**extensive**” (R. uEB3) and “**informative**” (R. Kzh4), “**clearly demonstrating the contributions of each key module**” (R. xAXU, R. Kzh4).

In response to the reviewers’ constructive suggestions, we acknowledge that our work can be further improved and have addressed all concerns as follows:

- **For clarity**, we have provided more details about the construction of training and evaluation data (suggested by R. Kzh4 and R. Sdzw), added more explanations to Figure 6 to avoid possible misunderstanding (by R. xAXU), and fixed a bug in writing (suggested by R. Kzh4).
- **For related work**, we have added a discussion of the difference between our work and video editing (suggested by R. Kzh4).
- **For design choice**, we added an experimental discussion of **motion representation** (suggested by R. uEB3) and a new ablation study on **different input types** (image vs. mask) to further validate our design choice (suggested by R. Kzh4).
- **For experiments**, we have added more successful results on additional interaction scenarios, including **cross-domain transfer** (suggested by R. xAXU), **two-hand coordination** and **long-horizon generation** (suggested by R. Sdzw). We also provided more **failure cases** in extremely challenging scenarios (suggested by R. xAXU, R. Sdzw).

We believe our work provides an effective solution for manipulation video generation. We hope these contributions will inspire further advances in controllable video synthesis and embodied intelligence. Thank you again for your valuable time and feedback.

Kind regards,

*All the authors*

---

### Meta-Review · Area_Chair_vLmj · 2025-12-29

**Summary:**

The paper proposes MIMIC, a two-stage image-to-video diffusion framework designed to generate controllable manipulation videos for embodied AI. It addresses the challenge of data scarcity by using reference videos to guide the generation of semantic interaction masks (Stage I) and subsequently using a Pair Prompt Control mechanism to disentangle object motion from camera motion for high-fidelity video synthesis (Stage II).

This paper was reviewed by four experts in the field. The recommendations are mixed, with scores of 8, 6, 4, 4.
The reviewers like the novel two-stage design (IMA and PPC modules) and recognized the paper's potential impact on addressing data scarcity for embodied intelligence (Sdzw, uEB3). They appreciated the clear disentanglement of camera and object motion(xAXU) and the extensive ablation studies validating the design choices (Kzh4, uEB3).

The reviewers raised the following concerns:
- doubts about cross-domain generalization and whether the motion is simply copied without physical understanding (xAXU)
- lack of clarity on data construction and the choice of motion representation (masks vs. flow) (Kzh4, uEB3)
- limitations regarding reference video dependency (Sdzw)

After rebuttal, most concerns are well solved.

**Reviewer Concerns:**

**Well addressed:**
- Generalization capabilities: The authors added cross-domain experiments (human hand to robot gripper) and failure case analysis, addressing concerns from Reviewer xAXU and Kzh4.
- Mechanism clarification (Copy vs. Generation): The authors explained the motion vector differences in Figure 6, proving the model adapts to target constraints rather than just copying reference pixels (xAXU).
- Data construction details: The explanation of manipulation templates and structured prompts clarified the training pair formation (Kzh4).

**Partly addressed:**
- Base Model Selection: The authors acknowledged the base model (DynamiCrafter) is slightly dated compared to closed-source SOTA but argued the method is transferrable. This is a reasonable research constraint but remains a limitation of the current implementation (uEB3).

**Unsolved:**

None

**Reviewer Scores:**

**Sdzw (8)**

The reviewer was already positive and the additional experiments on long-horizon generation further addresses the concerns. Score likely remains 8.

**uEB3 (6)**

The response regarding motion representation (masks vs. points) was logical, citing robustness. The score likely remains 6.

**xAXU (4)**

This reviewer questioned if the method was just copying motion. The rebuttal showed clear evidence (vector fields) that it is not. The cross-domain experiments also addressed their main weakness. The score likely remains 4 or slightly increases.

**Kzh4 (4)**

The reviewer's main issues were clarity (data pairs, stage I rationale). The revisions in the appendix and the explanation of context understanding for Stage I directly answered these. The score likely increases to 6.

---

### Decision · Program_Chairs · 2026-01-26

Accept (Poster)